



# Multi-generation OH oxidation as a source for highly oxygenated organic molecules from aromatics

Olga Garmash[1,2], Matti P. Rissanen[1,2], Iida Pullinen[3,9], Sebastian Schmitt[3,10], Oskari Kausiala[1,11],
Ralf Tillmann[3], Carl Percival[4], Thomas J. Bannan[4], Michael Priestley[4,5], Åsa M. Hallquist[6],
Einhard Kleist[7], Astrid Kiendler-Scharr[3], Mattias Hallquist[5], Torsten Berndt[8], Gordon McFiggans[4],
Jürgen Wildt[3,7], Thomas Mentel[3], and Mikael Ehn[1]

[1]Institute for Atmospheric and Earth System Research / Physics, Faculty of Science, University of Helsinki, Helsinki, Finland
[2]Aerosol Physics Laboratory, Physics Unit, Faculty of Engineering and Natural Sciences, Tampere University, Tampere, Finland
[3]Institut für Energie- und Klimaforschung, IEK-8: Troposphäre, Forschungszentrum Jülich GmbH, Jülich, Germany
[4]School of Earth and Environmental Sciences, University of Manchester, Manchester, UK
[5]Atmospheric Science, Department of Chemistry and Molecular Biology, University of Gothenburg, Gothenburg, Sweden
[6]IVL Swedish Environmental Research Institute, Gothenburg, Sweden
[7]Institut für Bio- und Geowissenschaften, IBG-2: Pflanzenwissenschaften, Forschungszentrum Jülich GmbH, Jülich, Germany
[8]Leibniz-Institut für Troposphärenforschung (TROPOS), Permoserstraße 15, 04318 Leipzig, Germany
[9]Present address: Department of Applied Physics, University of Eastern Finland, Kuopio, Finland
[10]Present address: TSI GmbH, Aachen, Germany
[11]Present address: Kärsa Oy, Helsinki, Finland

**Correspondence:** Olga Garmash (olga.garmash@helsinki.fi) and Mikael Ehn (mikael.ehn@helsinki.fi)

**Abstract.**

Recent studies have recognized highly oxygenated organic molecules (HOM) in the atmosphere as important in the formation of secondary organic aerosol (SOA). A large number of studies have focused on HOM formation from oxidation of biogenically emitted monoterpenes. However, HOM formation from anthropogenic vapours has so far received much less attention. Previous studies have identified the importance of aromatic volatile organic compounds (VOC) for SOA formation. In this study, we investigated several aromatic compounds, benzene ($C_6H_6$), toluene ($C_7H_8$), and naphthalene ($C_{10}H_8$), for their potential to form HOM upon reaction with hydroxyl radicals (OH). We performed flow tube experiments with all three VOC, and focused in detail on benzene HOM formation in the Jülich Plant Atmosphere Chamber (JPAC). In JPAC, we also investigated the response of HOM to $NO_X$ and seed aerosol. Using a nitrate-based chemical ionization mass spectrometer (CI-APi-TOF), we observed the formation of HOM in the flow reactor oxidation of benzene from the first OH attack. However, in the oxidation of toluene and naphthalene, which were injected at lower concentrations, multi-generation OH oxidation seemed to impact the HOM composition. We tested this in more detail for the benzene system in the JPAC, which allowed for studying longer residence times. The results showed that the apparent molar benzene HOM yield under our experimental conditions varied from 4.1 to 14.0%, with a strong dependence on the OH concentration, indicating that the majority of observed HOM formed through multiple OH-oxidation steps. The composition of the identified HOM in the mass spectrum also supported this hypothesis. By injecting only phenol into the chamber, we found that phenol oxidation cannot be solely responsible for the observed



HOM in benzene experiments. When $NO_X$ was added to the chamber, HOM composition changed and many oxygenated nitrogen-containing products were observed in CI-APi-TOF. Upon seed aerosol injection, the HOM loss rate was higher than predicted by irreversible condensation, suggesting that some undetected oxygenated intermediates also condensed onto seed aerosol, which is in line with the hypothesis of multi-generation HOM. Based on our results that HOM yield and composition

in aromatic systems strongly depend on OH and VOC concentration, we conclude that atmospheric models should account for such dependency and the chemical regime when implementing the quantitative results of laboratory studies. We also suggest that the dependence of HOM yield on chamber conditions may explain part of the variability in SOA yields reported in the literature.

**1   Introduction**

Highly oxygenated organic molecules (HOM) have been identified as large contributors to atmospheric secondary organic aerosol (SOA) in forested environments (Ehn et al., 2014; Öström et al., 2017; Bianchi et al., 2019). HOM form through a process called autoxidation, where intramolecular hydrogen shifts in organic peroxy radicals are followed by addition of molecular oxygen (Crounse et al., 2013), causing a rapid increase in the oxygen content of the molecules. The product is a new

peroxy radical, with an additional hydroperoxide functionality, that may be able to experience additional H-shifts. A wealth of studies have shown that this process is especially efficient in the oxidation of molecules with endocyclic double bonds (e.g., Rissanen et al., 2014; Jokinen et al., 2014; Mentel et al., 2015; Berndt et al., 2015; Rissanen et al., 2015; Berndt et al., 2016), a feature typical of biogenically emitted volatile organic compounds (VOC) such as monoterpenes.

While the formation pathways of HOM from biogenic VOC as well as their impact on atmospheric aerosol formation

has been studied extensively over the past years, the potential of anthropogenic VOC to form HOM has received much less attention. Wang et al. (2017) showed both computationally and experimentally that the yield of HOM from the hydroxyl radical (OH) initiated oxidation of alkyl benzenes increased with the size of the alkyl group. A second study investigated HOM formation from OH oxidation of seven different aromatics, finding HOM yields mainly within 0.1-1% for single-ring aromatics, and a few percent for two polycyclic aromatics, naphthalene and biphenyl (Molteni et al., 2018). These yields are

comparable in magnitude with those reported from ozone and OH oxidation of monoterpenes (e.g., Jokinen et al., 2015; Berndt et al., 2016). As aromatics are thought to be the most efficient precursors of SOA in urban areas (Kroll and Seinfeld, 2008), further studies of HOM formation, as well as their contribution to SOA, are necessary.

The most abundant aromatics in the atmosphere are benzene and alkylated benzenes, i.e. toluene, xylenes and trimethylbenzenes. Their primary sources are traffic, fuel handling and industrial processes. Aromatic compounds can constitute up to 20%

of urban VOC (Calvert et al., 2002) and in extremely polluted environments, such as next to a road with heavy traffic, their total concentrations can reach up to tens of ppb (Liu et al., 2008). In addition, vegetation also emits a wide range of aromatic





compounds, often in oxygenated form, and the total amount of the potential emissions may even match the anthropogenic sources (Misztal et al., 2015).

The major sink of aromatics in the atmosphere is the reaction with OH (Atkinson and Arey, 2003), which in most cases involves OH addition to the aromatic ring and the formation of a carbon-centred radical. In the case of benzene, more than half

of these radicals will end up forming phenol (Volkamer et al., 2002; Berndt and Böge, 2006). The remainder of the products can undergo $O_2$-additions and isomerisation, forming bicyclic peroxy radicals (BPR) or result in epoxides (Bloss et al., 2005; Glowacki et al., 2009; Wang et al., 2013). As suggested by Molteni et al. (2018), the BPR may undergo further autoxidation to form HOM. However, the produced phenol will be abundant, which upon reaction with OH can also produce a BPR with low yield, about 10% (Master Chemical Mechanism, MCMv3.3.1, Bloss et al., 2005). The reaction rate coefficient of phenol with

OH is about 20 times higher than that of benzene, meaning that we cannot ignore its role in the total HOM formation following further oxidation steps. For instance, Schwantes et al. (2017) showed that methylphenol (cresol) formed in toluene oxidation was a much more important SOA precursor than its branching ratio (20%) would suggest. In their study, only a minor fraction of the identified compounds would classify as HOM according to the definition suggested by Bianchi et al. (2019), that six or more O-atoms are required for a molecule to be classified as HOM. However, the authors demonstrated the importance of

multiple OH oxidation steps for SOA formation.

Several studies over the last decades examined the SOA yields from oxidation of aromatics, with disparate results that remain largely unexplained. The suggested causes are the differences in the exact experimental conditions (Ng et al., 2007; Hildebrandt et al., 2009; Emanuelsson et al., 2013). These include differences in VOC loading, UV light intensity, and the concentration of $NO_X$ ($NO+NO_2$). Being a by-product of combustion, $NO_X$ is on a large scale co-emitted with aromatic VOC. $NO_X$, and

especially NO, will decrease the lifetime of $RO_2$ radicals in the atmosphere, in direct competition with autoxidation (Praske et al., 2018; Rissanen, 2018). Additionally, highly oxygenated $RO_2$ radicals can combine efficiently to form ROOR' dimers (Berndt et al., 2018a, b). These dimers are often the least volatile oxidation products, with a particularly large influence on the formation of new particles (e.g., Tröstl et al., 2016; Lehtipalo et al., 2018), but under high-$NO_X$ conditions their formation becomes supressed (e.g., Ehn et al., 2014; Rissanen, 2018) .

The measurement of HOM relies mainly on the use of the chemical ionisation atmospheric pressure interface time-of-flight mass spectrometer, CI-APi-TOF (Jokinen et al., 2012). In combination with wall-less CI inlet, nitrate ion ionisation is typically used due to its selectivity towards molecules with several H-bond donors, such as the multi-hydroperoxides typically formed in autoxidation (Hyttinen et al., 2015). Until now, the application of the CI-APi-TOF to measuring HOM from aromatics has been limited to a few studies (Wang et al., 2017; Molteni et al., 2018), and these have been performed in flow reactors with

residence times of 20 seconds or less. To understand the importance of aromatic-derived HOM in the atmosphere, systematic studies, including experiments at varying conditions and longer timescales, are needed.

In this study, we investigated the OH-initiated oxidation of aromatics, with a strong focus on benzene. We conducted experiments in a flow reactor and a continuously stirred tank reactor (JPAC) in order to determine HOM composition and yield over a wide range of conditions. In the JPAC runs, we varied both VOC and OH concentrations, and tested the influence of





NO$_X$ on the HOM distribution. Benzene was also substituted by phenol in order to test different oxidation pathways. Finally, we explored the contribution of HOM to SOA formation by adding seed aerosol.

## 2    Aromatic Oxidation Chemistry

In this section, we outline the relevant oxidation steps of aromatic compounds with a focus on benzene. In oxidation reactions initiated by OH, the oxidation propagation and termination will determine the chemical composition of the product molecules. These reactions will change the amount of hydrogen (H), carbon (C), oxygen (O) and nitrogen (N) atoms in the detected oxidised species and are therefore central to our discussion. In this section, we do not attempt to review all of the existing studies. Instead, we present an overview of relevant products and radicals formed in benzene oxidation by OH. We also discuss the relevant chain propagating and terminating reactions of organic peroxy radicals (RO$_2$) as the main intermediates of gas-phase oxidation. Detailed mechanistic descriptions of benzene oxidation can be found in the literature (Calvert et al., 2002; Volkamer et al., 2002; Bloss et al., 2005; Glowacki et al., 2009; Wang et al., 2013; Vereecken, 2019, and references therein).

### 2.1    Oxidation by OH

Benzene (C$_6$H$_6$) oxidation by OH almost exclusively initiates via addition of OH to the aromatic ring (Glowacki and Pilling, 2010; Bloss et al., 2005), while abstraction of H-atoms from the ring is a minor pathway. The addition of OH creates a carbon-centred radical C$_6$H$_7$O•. According to previous studies, about 53 - 61% of these radicals will form phenol, where the aromatic ring is retained and C$_6$H$_6$O molecule has one OH group (one more O atom) (Volkamer et al., 2002; Berndt and Böge, 2006). The remaining fraction of C$_6$H$_7$O• will add molecular oxygen (O$_2$) forming a C$_6$H$_7$O$_3$• peroxy radical (Lay et al., 1996). This RO$_2$ can undergo endo-cyclization, where RO$_2$ attacks its own double bond to form an oxygen bridge resulting in an alkyl radical. This radical then reacts again with O$_2$ and forms a bicyclic peroxy radical (BPR) C$_6$H$_7$O$_5$• (Glowacki et al., 2009; Birdsall et al., 2010; Wang et al., 2013). In this pathway OH attachment and addition of two O$_2$ increases the molecular composition of parent benzene by five O atoms (and one H), and subsequent reactions generally lead to radical termination, and potential molecular fragmentation (Jenkin et al., 2003; Wang et al., 2013). Studies have also reported a minor channel in which the C$_6$H$_7$O$_3$• bicyclic alkyl radical isomerises and forms an epoxide functionality, though the importance of this pathway under atmospheric conditions is yet unclear (Bloss et al., 2005; Glowacki et al., 2009; Wang et al., 2013).

For substituted aromatics, the set of reactions is similar to that described above, though branching ratios are different (Birdsall and Elrod, 2011). For instance, in toluene oxidation by OH, the BPR forms with about 65% yield, which is about twice that formed in the case of benzene (MCMv3.3.1, Bloss et al., 2005). In addition, the presence of methyl groups increases the chances of H-abstraction by OH radicals as well as increases the OH-VOC reaction rate coefficient (k$_{OH}$) (Bloss et al., 2005; Atkinson, 1994a, b; Atkinson and Aschmann, 1989).

A distinct feature of aromatic oxidation is the faster oxidation rates of first-generation products as compared to the parent molecule. For instance, benzene has a k$_{OH}$ of $1.22 \times 10^{-12}$ cm$^3$ s$^{-1}$ at 298 K, while k$_{OH}$ of phenol is about 20 times larger, $2.82 \times 10^{-12}$ cm$^3$ s$^{-1}$, and k$_{OH}$ of catechol (a primary product of phenol oxidation) is about 100 times higher ($1.0 \times 10^{-10}$ cm$^3$ s$^{-1}$)





(Atkinson et al., 2006). In the case of less thoroughly investigated oxidation products, $k_{OH}$ is likely to increase in comparison to benzene itself, as the pi-electron structure of benzene makes it less susceptible towards OH oxidation compared to most organic molecules. The process of sequential oxidation is commonly known as ageing and in general should lead to eventual fragmentation of the products retaining in the gas phase (Chacon-Madrid and Donahue, 2011).

## 2.2 RO$_2$ radical reactions

### 2.2.1 Chain propagation

Chain propagation refers to the reactions that result in another radical (i.e., still has an unpaired electron). These reactions can be bimolecular, happening upon collision with another molecule, or unimolecular, occurring within the molecule. The reaction rates depend on the structure of the compound as well as the concentration of potential bimolecular reaction partners.

A bimolecular propagation reaction proceeds through formation of alkoxy radicals (RO) when an RO$_2$ radical reacts with another R'O$_2$ (forming RO+R'O+O$_2$) or NO (forming RO+NO$_2$). This reaction decreases the oxygen content per molecule by one and is one of the most common reactions for peroxy radicals (Finlayson-Pitts and Pitts, 2000). The fate of the RO radicals depends on their structure. They can decompose, undergo H-shifts, or react with O$_2$. In case of benzene, decomposition of alkoxy radicals may lead to ring scission and potentially further autoxidation. However, in case of first-generation BPR from benzene, the Master Chemical Mechanism (MCMv3.3.1) predicts that BPR will react with HO$_2$ or RO$_2$ forming RO radicals with branching ratios of 23% and 60%, respectively, and eventually decompose into smaller molecules (Jenkin et al., 2003; Bloss et al., 2005).

Autoxidation of RO$_2$ radicals is one important reaction chain recently shown in oxidation of monoterpenes and other alkenes (Crounse et al., 2013; Ehn et al., 2014). It involves intramolecular hydrogen shifts to the peroxide group from other carbon atoms and subsequent addition of oxygen to the produced carbon-centred radicals. While autoxidation involves both uni- and bimolecular reactions, the high abundance of oxygen in the air allows autoxidation to be pseudo-unimolecular (Rissanen et al., 2015). Autoxidation is more likely to happen at lower RO$_2$ concentrations and for RO$_2$ with larger amount of functional groups (Crounse et al., 2013). It may also occur in aromatic molecules following the initial bicyclic peroxy radical, i.e. $C_6H_7O_5\bullet$ in case of benzene (Wang et al., 2017). H-shift itself does not modify the molecular composition, but O$_2$ additions increase the oxygen content by an even number. Autoxidation can proceed until the H-shift potential is exhausted and, at least in monoterpene systems, can often be competitive with bimolecular termination reactions at atmospheric conditions (see next section). However, autoxidation for aromatic compounds is not yet well understood, and until recently, the bicyclic peroxy radical was considered the most oxygenated first-generation product.

Other propagation reactions mostly include fragmentation. During autoxidation, H atoms may be abstracted from a terminal carbonyl group creating an acyl radicals (RC$\bullet$=O), which may eliminate a CO from the molecule and leave an alkyl radical where further O$_2$ can attach. In this reaction, one C atom and one O atom are lost (Crounse et al., 2012; Rissanen et al., 2014). If CO is not eliminated, O$_2$ will add and, upon a reaction with another RO$_2$ (or NO), an RO radical will split CO$_2$, losing one C and two O atoms instead (Orlando et al., 2003; Atkinson and Arey, 2003; Vereecken and Peeters, 2009).





### 2.2.2 Chain termination

Termination reactions proceed in competition with the chain propagation reactions described above. Termination reactions result in "closed-shell" molecules containing only paired electrons. An example of a unimolecular termination process is the ejection of OH following an H-abstraction from a carbon with a hydroperoxide group, forming a carbonyl (C=O), meaning a

loss of one H and one O atom (Rissanen et al., 2014).

A number of bimolecular termination reactions can take place. First, $RO_2$ can react with $HO_2$, and form ROOH hydroperoxides, which adds one H atom (as $O_2$ is ejected). Alternatively, $RO_2$ can react with another $R'O_2$ and form a dimer ROOR' and $O_2$, where the number of C and H atoms of $RO_2$ and $R'O_2$ in sum are conserved, while two O-atoms are lost. $RO_2$ can also upon collision with $R'O_2$ form an alcohol (ROH) or an aldehyde (RCHO) and $O_2$ in which the molecule will have lost an oxygen

and either gained or lost a hydrogen, compared to the initial $RO_2$ (Finlayson-Pitts and Pitts, 2000). In addition, RO radicals, mentioned in the previous section, may terminate upon reaction with $O_2$ forming a carbonyl compound with one less H atom.

In the atmosphere, NO and $NO_2$ can be effective in terminating $RO_2$, although the major reaction between $RO_2$ and NO is chain-propagating to form $NO_2$ and RO. NO can add to $RO_2$ to form organonitrates while $NO_2$ upon reaction with $RO_2$ can form thermally unstable peroxynitrates ($RO_2NO_2$) or more stable peroxyacylnitrates $RO_2(O)NO_2$ (Zabel, 1995; Atkinson,

2000; Orlando and Tyndall, 2012; Rissanen, 2018). In the case of aromatics, RO can also be long-lived enough to react with $NO_2$ to form nitrophenol-type compounds (Olariu et al., 2002; Jenkin et al., 2003; Bloss et al., 2005). NO and $NO_2$ addition to the molecule consequently changes its composition, and are easy to identify based on the added N-atom. However, separating between nitrogen-containing HOM with nitro-, nitrate- or peroxynitrate functionalities is impossible with our instrumentation, and can only be speculated based on experimental conditions like the $NO/NO_2$ ratio.

## 20   3   Methods

### 3.1   Experimental set-up

The gas-phase oxidation experiments were conducted in two different laboratory settings. Initial experiments were performed in a flow reactor at the University of Helsinki, focusing on determination of HOM distributions during the oxidation of benzene, toluene and naphthalene. The flow reactor allowed fast oxidation experiments at high VOC concentrations providing

a possibility for comparison with previous studies. Motivated by the findings in the flow reactor, we performed subsequent quantitative studies on HOM formation from benzene oxidation at the Jülich Plant Atmosphere Chamber facility (JPAC) at Research Centre Jülich (Mentel et al., 2009). Using JPAC allowed us to do experiments at longer time scales and more varied experimental conditions. In the following sub-sections, the two facilities are described in more detail, as are the types of experiments conducted in each of them.





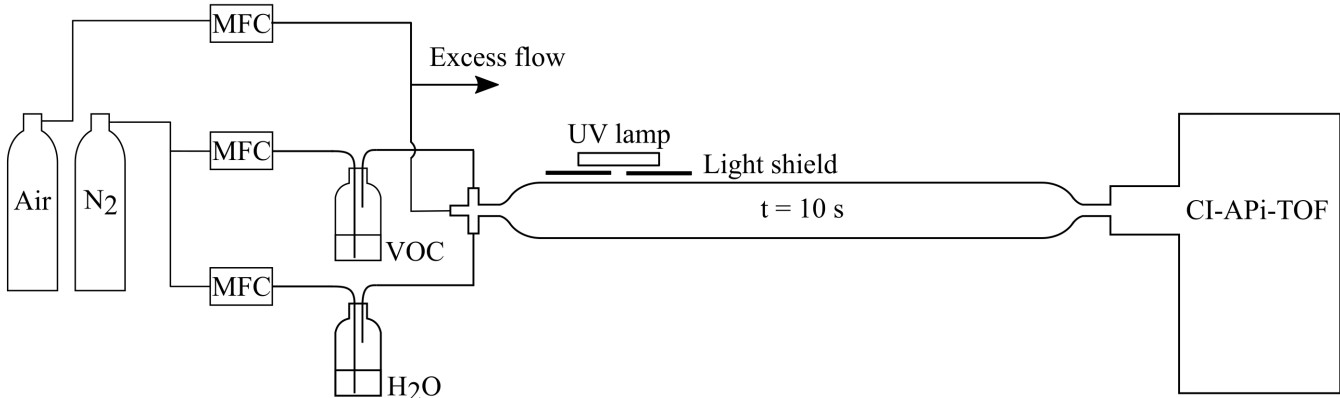

**Figure 1.** Flow reactor set-up used at the University of Helsinki. The aromatic VOC and water vapour were mixed with synthetic air at the inlet of the flow reactor. A shielded UV lamp irradiated a small part of the reactor, forming OH radicals by photolysing the water vapour. The inlet flow to the CI-APi-TOF mass spectrometer (see Sect. 3.2.1) defined the residence time (~10 seconds) in the reactor.

### 3.1.1   University of Helsinki flow reactor

The flow reactor utilised in this work was made of quartz and had a volume of 2 litres. At the inlet of the flow reactor, reactant gases were mixed in a Swagelok steel cross. We operated the system at room temperature (~22 °C) with high VOC concentrations. Synthetic air (80% $N_2$/20% $O_2$; AGA purity 5.0, 99.999%) was used as the main carrier gas, while VOC and water were added via separate lines by bubbling nitrogen (cryogenic $N_2$, AGA) through vials containing the respective liquids (see Fig. 1 for a schematic depiction of the set-up). In the case of the solid naphthalene, nitrogen flow was passed over granules of the compound. An ultraviolet (UV) lamp was attached on the top of the reactor, irradiating a small part of it through a light shield. Hydroxyl radicals were produced via photolysis of water at 184.9nm. The total flow through the flow reactor was 12 standard litres per minute (slpm), leading to a 10 second residence time inside the reactor.

In the flow reactor experiments, uncertainties in the VOC and OH concentrations were large enough that no quantitative analysis was attempted, but instead we focused on the chemical composition of the HOM products. We could only roughly approximate the VOC concentrations in the flow reactor. Assuming the flow over the VOC in the vial (0.01-0.05 slpm) was saturated, we got the following estimated concentrations in the flow reactor for different experiments: benzene ~400 ppm, toluene ~25 ppm, and naphthalene ~0.4 ppm. The photolysis rate could not be determined in the present set-up, and thus no attempt was made to calculate OH concentrations in the flow reactor. We also conducted direct VOC photolysis experiment in absence of water to determine the effect of this process on the product spectrum.



### 3.1.2   Jülich Plant Atmosphere Chamber (JPAC)

In this study, we used the larger chamber of the JPAC facility (1450 l), made from Borosilicate glass (Mentel et al., 2009). It was operated as a continuously stirred tank reactor with modifications as described in Mentel et al. (2015). The chamber was positioned in a temperature-controlled housing and the temperature throughout the experiments was kept at 14.2 ± 0.3 °C.

Purified air was fed into the system at a flow rate of ~30 l min⁻¹ allowing a ~48 minute residence time. A slight overpressure of 5 mbar was maintained to reduce the leaking of ambient air into the chamber. Inflow to the chamber was from two separate lines, one used to feed ozone and humidified air, the other to introduce VOC and $NO_X$ mixed into dry air. The RH in the chamber was maintained at 65 ± 3%.

Benzene was fed into the chamber from a diffusion source with a constant flow, and the concentration in the chamber could
be varied according to what fraction of this flow was diverted into the chamber. The procedure was identical in the experiments where phenol was used instead of benzene. OH radicals were produced by ozone photolysis in presence of water vapour. The UV lamp (Philips, TUV 40W, $\lambda_{max}$ = 254nm) was located inside the chamber and was shielded from both ends with UV-absorbing glass tubes. OH production could be varied by either adjusting the concentration of ozone or the light intensity by changing the size of the gap between the UV-absorbing tubes on the UV lamp. Starting ozone concentration was varied
between 15 and 100 ppb resulting in OH concentrations $1.2 – 45 \times 10^7$ cm⁻³. These parameters, together with the concentration of benzene, determined the final concentration of OH inside the chamber.

The influence of $NO_X$ on the benzene oxidation system was studied by injecting NO into the chamber. The injected NO resulted in 4.3 ppb of $NO_X$. UV-A lights (12 x HQI, 400 W/D; Osram, Munich, Germany) around the chamber were used to photolyse $NO_2$ to NO and O, the latter reacting with $O_2$ to form ozone. Ozone consequently reacts with NO to reform $NO_2$. A
photostationary state with a constant $NO_2$:NO ratio of roughly 3:1 was achieved at a given ozone concentration (~58 ppb) and photolysis rate ($J_{NO_2}$, ~4.2x10⁻³ s⁻¹).

In certain experiments, monodisperse 100 nm seed aerosol particles consisting of dry ammonium sulfate (($NH_4$)$_2SO_4$, AS) were introduced into the chamber. The AS particles were formed by atomising an ammonium sulfate water solution, which were then dried using silica gel and size-selected using a differential mobility classifier (TSI Inc, 3071). Before particles were
added, pure water was nebulised to ensure a constant flow into the chamber. The achieved aerosol had a bimodal distribution, as ~25% of the particles were doubly charged particles of larger size, which, having the same electrical mobility, entered the chamber. This was considered when calculating the condensational sink in the chamber (CS). The seed addition experiments helped assessing the amount of SOA that was formed from low-volatile compounds, as the increased CS shifted their main sink from the chamber walls to the aerosol. The method is described in more detail by Ehn et al. (2014).
In this work, we utilize a total of 27 benzene + OH experiments, 3 phenol + OH experiments, 1 benzene + OH + NOx experiment and 1 seed-addition experiment (Table 1). The reaction of benzene and ozone under dark conditions as well as photolysis in absence of ozone were also tested. In these tests, no HOM were detected, and we thus attribute the VOC + OH reaction to be the initiator of all measured HOM in this work. The parameters for each experiment were determined when the chamber had reached steady-state. Typically, each experiment started by adjusting VOC and $O_3$ concentrations, after which the





**Table 1.** Summary of the experimental parameters from the JPAC chamber. QMS/TOF refers to quadruple / time-of-flight detector in the proton transfer reaction (PTR) mass spectrometer used for measuring VOC.

**benzene**

| # | PTR | UV lamp, J(O$^1$D), $10^{-3}$ s$^{-1}$ | VOC, ppb | OH, $10^7$ cm$^{-3}$ | HOM, $10^7$ cm-3 | CS, s$^{-1}$ | HOM yield, % |
|---|-----|------|------|------|------|------|------|
| 1 | QMS | 2.6 | 4.5 | 25.4 | 30.6 | $1.3 \times 10^{-3}$ | 10.9 |
| 2 | QMS | 2.6 | 6.4 | 8.7 | 12.2 | $6.8 \times 10^{-6}$ | 8 |
| 3 | QMS | 2.6 | 6.5 | 8.5 | 12.1 | $6.3 \times 10^{-6}$ | 8 |
| 4 | QMS | 2.6 | 5.3 | 16.5 | 23.7 | $4.5 \times 10^{-4}$ | 10.2 |
| 5 | QMS | 2.6 | 4.1 | 29.5 | 34 | $4.2 \times 10^{-3}$ | 14 |
| 6 | QMS | 1.5 | 4.4 | 20.3 | 24.9 | $2.7 \times 10^{-4}$ | 10.4 |
| 7 | QMS | 2.6 | 3.7 | 29.1 | 30.8 | $1.2 \times 10^{-3}$ | 11.6 |
| 8 | QMS | 4.9 | 2.9 | 44.6 | 36.3 | $3.3 \times 10^{-3}$ | 13.2 |
| 9 | QMS | 3.8 | 3.1 | 42.6 | 36.2 | $1.9 \times 10^{-3}$ | 11.5 |
| 10 | QMS | 2.6 | 6.3 | 6.1 | 8.4 | $8.0 \times 10^{-6}$ | 8 |
| 11 | QMS | 2.6 | 2.5 | 6.1 | 4.6 | $7.2 \times 10^{-6}$ | 11 |
| 12 | TOF | 2.6 | 13.3 | 3.6 | 10.3 | $1.3 \times 10^{-5}$ | 7.9 |
| 13 | TOF | 2.6 | 30.4 | 2.6 | 13.7 | $1.6 \times 10^{-5}$ | 6.5 |
| 14 | TOF | 2.6 | 2.4 | 7.4 | 4.6 | $9.8 \times 10^{-6}$ | 9.4 |
| 15 | TOF | 2.6 | 1.6 | 10.6 | 4.1 | $1.0 \times 10^{-4}$ | 8.8 |
| 16 | TOF | 2.6 | 6.9 | 6 | 9.5 | $9.0 \times 10^{-6}$ | 8.4 |
| 17 | QMS | 2.6 | 112.4 | 1.2 | 19.5 | $4.9 \times 10^{-5}$ | 5.2 |
| 18 | QMS | 4.9 | 105.7 | 3.3 | 30.9 | $6.8 \times 10^{-3}$ | 5.3 |
| 19 | TOF | 4.9 | 95.1 | 4.7 | 30.6 | $2.0 \times 10^{-2}$ | 6.9 |
| 20 | TOF | 2.6 | 14.9 | 10.2 | 26 | $4.1 \times 10^{-3}$ | 8.5 |
| 21 | TOF | 2.6 | 14.5 | 10 | 26.3 | $4.0 \times 10^{-3}$ | 9 |
| 22 | TOF | 2.6 | 75.3 | 4 | 29.7 | $3.8 \times 10^{-3}$ | 4.8 |
| 23 | TOF | 2.6 | 94.9 | 2.9 | 30.8 | $2.8 \times 10^{-3}$ | 5 |
| 24 | TOF | 2.6 | 64.9 | 3.2 | 30.6 | $2.3 \times 10^{-3}$ | 6.4 |
| 25 | TOF | 2.6 | 84.1 | 2.6 | 32.1 | $2.1 \times 10^{-3}$ | 6.4 |
| 26 | QMS | 2.6 | 15.7 | 4.1 | 7.2 | $9.4 \times 10^{-5}$ | 4.1 |
| 27 | TOF | 1.7 | 16.1 | 5.9 | 22.2 | $3.2 \times 10^{-4}$ | 8.8 |





| phenol | | | | | | | |
|---|---|---|---|---|---|---|---|
| # | PTR | UV lamp, $J(O^1D)$, $10^{-3}$ $s^{-1}$ | VOC, ppb | OH, $10^7$ $cm^{-3}$ | HOM, $10^7$ cm-3 | CS, $s^{-1}$ | HOM yield, % |
| 1 | TOF | 2.6 | 5.4 | 2.2 | 19.7 | $1.7x10^{-3}$ | 2.5 |
| 2 | TOF | 3.8 | 4.6 | 2.7 | 21.2 | $3.5x10^{-3}$ | 3 |
| 3 | TOF | 1.7 | 7.1 | 1.4 | 17 | $3.4x10^{-4}$ | 2.3 |

| Benzene+NOx | | | | | | | | |
|---|---|---|---|---|---|---|---|---|
| # | PTR | UV lamp, $J(O^1D)$, $10^{-3}$ $s^{-1}$ | VOC, ppb | OH, $10^7$ $cm^{-3}$ | HOM, $10^7$ cm-3 | CS, $s^{-1}$ | HOM yield, % | NOx, ppb (NO2:NO) |
| 1 | TOF | 2.6 | 15.4 | 5.7 | - | - | - | 4.3 (3:1) |

| Aerosol seed experiment | | | | | | | | |
|---|---|---|---|---|---|---|---|---|
| # | PTR | UV lamp, $J(O^1D)$, $10^{-3}$ $s^{-1}$ | VOC, ppb | OH, $10^7$ $cm^{-3}$ | HOM, $10^7$ cm-3 | CS, $s^{-1}$ | HOM yield, % | SOA yield, % |
| 1 | QMS | 2.6 | 15.7 | 4.1 | 7.2 | $9.4x10^{-5}$ | 4.1 | 40 |

UV lamp was switched on. Especially at high VOC concentrations, this initiated a strong particle formation event, and it took several hours to reach a steady-state. An example experiment is presented in Fig. 2.

## 3.2 Instrumentation

### 3.2.1 CI-APi-TOF

5   A Chemical Ionisation Atmospheric Pressure interface Time-Of-Flight mass spectrometer (CI-APi-TOF, Jokinen et al., 2012) was used to measure HOM in the Helsinki flow reactor and at JPAC. It consists of a Chemical Ionisation inlet (CI, Airmodus Oy) and an APi-TOF online high resolution mass spectrometer (Junninen et al., 2010, Tofwerk AG/Aerodyne Research Inc.). The CI inlet is designed to minimize wall contact during sampling and utilizes a high sample flow rate of around 10 slpm. Inside the CI inlet, the sample air is co-axially merged with a sheath flow (~20 slpm) of filtered compressed air that contains
10  nitric acid and nitrate ions. The ions were produced by exposure to either a radioactive source (241 Am $\alpha$-emitter in JPAC) or soft X-rays (<9.5keV, Hamamatsu L9490 photoionizer in Helsinki flow reactor). Upon collisions with neutral nitric acid, the nitrate ions can form $(HNO_3)_{0-2}NO_3^-$ adducts, which are referred to as reagent ions. Using an electric field, the reagent ions are pushed into the sample flow and, after ~200 ms of interaction between sample molecules and reagent ions, guided into the APi-TOF through a critical orifice admitting 0.8 l min$^{-1}$.





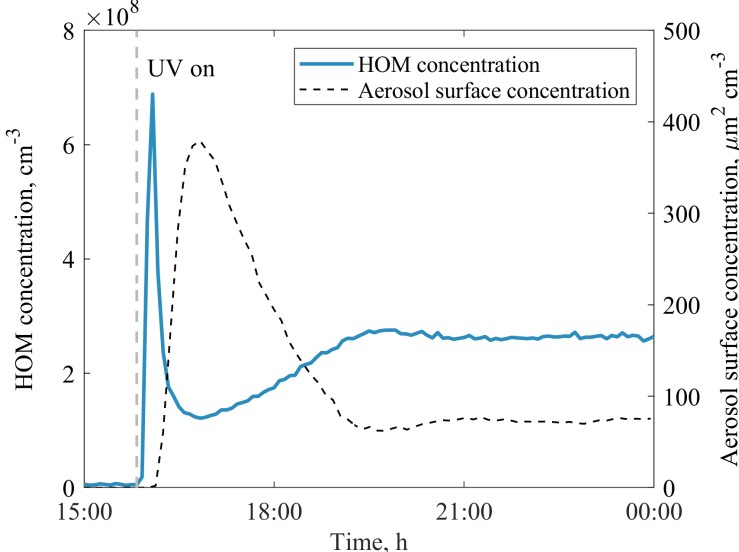

**Figure 2.** An example of JPAC reaching steady-state after the UV lamp was switched on in experiment #20. Within a few minutes, the concentration of HOM increased significantly triggering particle formation in the chamber, which acted as a sink for HOM. After both the gas- and particle-phase signals reached steady-state (in ~5 hours) the HOM yield was calculated.

If energetically favourable, molecules in the sample air can be ionised via proton transfer or adduct formation. In most cases, the ionization of a molecule M happens as

$$M + (HNO_3)_x \cdot NO_3^- \rightarrow M \cdot NO_3^- + (HNO_3)_x \tag{1}$$

In cases where M is a strong acid, such as sulfuric acid, it may transfer a proton to $NO_3^-$ and be detected in its deprotonated

form, but most molecules are detected in clusters with $NO_3^-$.

Nitrate is strongly bound to neutral nitric acid, and, therefore, ionisation through adduct formation will only happen with molecules that can compete for nitrate against neutral nitric acid (Hyttinen et al., 2015). As a result, the nitrate CI-APi-TOF is selective towards HOM, as they often contain two or more hydrogen bond donors in the form of –OOH (or –OH) groups, and are able to form stable adducts with the nitrate ion (Hyttinen et al., 2015).

Inside the atmospheric pressure interface (APi), sampled ions are guided through two differentially pumped quadrupoles and an ion lens assembly, in which the pressure is gradually decreased. After this, the ions enter the time-of-flight (TOF) chamber, where they are orthogonally extracted, and their flight time to the micro-channel plate (MCP) detector is measured. This flight time is converted to mass-to-charge ratio (Th) for each ion in data post processing. All data processing, including averaging, mass axis calibration and peak integration, was done using the tofTools software package for MATLAB (Junninen, 2013).

The molecular formulas of sampled ions could be resolved owing to the high resolution (~4000 Th/Th at 125 Th) of the TOF analyser. If an ion was identified to be a HOM, defined here simply as molecules with 6 or more O-atoms (Bianchi et al., 2019), and was the dominant ion (>80% of the signal) at its integer mass, the intensity was determined by integrating over the





whole integer mass where this HOM was observed. This approach was concluded to be the most robust method, as an accurate mass axis calibration was at times problematic to achieve, and at the 5-minute time resolution used, peak fitting uncertainties increased due to limited signal-to-noise ratio. By using this type of unit mass resolution (UMR) analysis, we avoided having small variations leading to signals "leaking" into closely lying ions that were also being fitted. While our approach does add

uncertainty to the quantification, it is believed to be on the order of 10% (as we limited ourselves to masses where the HOM was the dominant ion). This is much smaller than the uncertainty in the absolute sensitivity calibration of the CI-APi-TOF (see below). In addition, when determining the average HOM intensity for a particular experiment, the background signal, determined before the UV lights were switched on, was subtracted.

The HOM ion count rate was converted to concentration (molecules cm$^{-3}$) using the following equation (Jokinen et al.,
10   2012):

$$[HOM] = C_f \times \frac{\sum_i HOM_i \cdot NO_3^-}{\sum_{i=0}^{2}(HNO_3)_i \cdot NO_3^-} \qquad (2)$$

where HOM$_i$•NO$_3^-$ is the count rate of individual HOM clusters with NO$_3^-$ and the denominator describes the count rate of the reagent ions. C$_f$ is a calibration coefficient, which in the JPAC experiments in this work was approximated as $1.6 \times 10^{10}$ molecules cm$^{-3}$ following Ehn et al. (2014), who used gravimetric calibration with perfluoroheptanoic acid (PFHA) for the
same setup as used in this study. Ehn et al. (2014) reported the uncertainty of this method as ± 50%, and we estimate a slightly larger uncertainty here due to the lack of calibrations during our measurement campaign. We estimate an uncertainty in determination of the absolute concentration of ± 70% with the precision in relative changes of less than 10%.

### 3.2.2 PTR-QMS and PTR-TOF

VOC and their oxidation products in JPAC were measured by a high-sensitivity Proton-Transfer-Reaction Mass Spectrometer
(PTR-QMS, Ionicon Analytik GmbH). The technique is described by Lindinger et al. (1998). Calibrations of the VOC were performed using diffusion sources (Gautrois and Koppmann, 1999). The PTR-QMS operated at 2 minute time resolution and the sampling switched every 20 minutes between the inlet and the outlet of JPAC. The sampling lines consisted of ~10 meter long PFA tubing of 4 mm inner diameter and were heated to 60 °C. The sampling flow rate was 0.5 l min$^{-1}$. During part of the campaign, a high-resolution PTR-TOF, equipped with a time-of-flight mass spectrometer, was deployed (Graus et al.,
2010). The PTR-TOF was calibrated using an advanced Liquid Calibration Unit (LCUa, Ionicon Analytik GmbH) for phenol. Benzene calibrations were performed using a self-made compressed gas standard containing, among other VOC, also 670 ppb of benzene, further diluted using the LCUa. Sampling from the outlet of JPAC was performed via a 2 m long, 1 mm ID PEEK-sampling line heated to 60°C. In order to cover the VOC measurements during all experiments, the data from both instruments was used, giving preference to the PTR-TOF when it was available.

### 3.2.3 Aerosol Instrumentation

To measure the particle number size distribution in JPAC a scanning mobility particle sizer (SMPS, electrostatic classifier (TSI 3071) and condensation particle counter (TSI 3025), TSI Inc.) was deployed. The SMPS measured particle concentrations in





the size range from 14 to 820 nm in diameter, which were used to calculate the condensation sink (see Sect. 3.4). A high-resolution time-of-flight aerosol mass spectrometer (AMS, DeCarlo et al., 2006; Rubach, 2013, Aerodyne Research Inc.) was used to measure the composition of particles from ~40 nm to 1 $\mu$m in diameter in the JPAC. In the AMS, aerosol particles were vaporized at 600 ºC and ionized by electron impact ionization at 70 eV, after which the ions were guided via an ion lens into the
time-of-flight mass detector. The AMS was calibrated using ammonium nitrate particles and the concentration of ammonium sulfate and organic aerosol was determined by summing the corresponding fragment ions from the mass spectra. SOA yield was estimated from the AMS as the ratio of produced organic aerosol mass to consumed VOC.

### 3.3   Determination of OH concentration

In JPAC, the concentration of OH radicals in the experiment was calculated based on the amount of reacted VOC, for which
the reaction rate coefficient with OH is known. In the chamber, the concentration of a VOC unreactive to $O_3$ is represented by the following equation:

$$V \times \frac{d[VOC]}{dt} = F \times ([VOC]_{in} - [VOC]) - V \times k_1 \times [OH] \times [VOC] \tag{3}$$

where the V is the chamber volume, F is the flow rate through the chamber, $k_1$ is the reaction rate coefficient for OH with the VOC. $[VOC]_{in}$ indicates the average concentration of the precursor compound in the total flow entering the chamber, and
[VOC] and [OH] describe the actual concentrations in the chamber, whereby [VOC] is measured at the outflow of the chamber. During steady-state conditions, OH concentration in the chamber can be calculated as follows:

$$[OH] = \frac{1}{t \times k_1} \times \frac{[VOC]_{in} - [VOC]}{[VOC]} \tag{4}$$

where t=V/F is the residence time in the chamber. t was approximately constant throughout the campaign (2900 seconds) and $k_1$ for 14°C was taken as $1.19 \times 10^{-12}$ cm$^3$ s$^{-1}$ for benzene and $3.30 \times 10^{-11}$ cm$^3$ s$^{-1}$ for phenol (Bloss et al., 2005; Atkinson
and Aschmann, 1989). $[VOC]_{in}$ and [VOC] were both determined by PTR-QMS or PTR-TOF. This method is independent of the instrumental calibration; however, it assumes that benzene is lost solely through the reaction with OH. The determination of [OH] was verified in some experiments by introducing 1,8-cineole in addition to benzene, which confirmed the determined OH concentrations within 6-12%.

### 3.4   Determination of HOM yield in JPAC

We were able to calculate HOM molar yields from JPAC experiments. For HOM yields in this work, we take a slightly different approach than earlier studies where the yield has directly been equated with a branching ratio of a certain VOC-oxidant reaction. We define the molar yield $\gamma$ of HOM as the fraction of reacted VOC that produced HOM during the residence time in our chamber. This definition includes also HOM formation from molecules reacting multiple times with OH, i.e. multi-generation OH oxidation. We take this approach since the oxidation products will react with OH much more rapidly than the
parent VOC benzene, which subsequently means that the primary fate of the first generation oxidation products of benzene will be to undergo further OH reactions. In other words, the more atmospherically relevant quantity, for instance relating to SOA





formation, is the ultimate amount of HOM formed, rather than only the HOM branching ratio in the initial OH reaction. The change in HOM concentration in time is defined as HOM production rate minus HOM loss rate:

$$\frac{d[HOM]}{dt} = Production_{HOM} - Loss_{HOM} = k_1\gamma[VOC][OH] - k_{loss}[HOM]$$

where $k_1$ is the benzene-OH reaction rate coefficient, $\gamma$ is a HOM molar yield, and kloss is the total loss coefficient of HOM to

the chamber walls ($k_{wall}$) and to aerosol particles present in the chamber (i.e. the condensation sink, CS). Here we assume that HOM are low-volatile enough that these are the dominant loss pathways, and that flushout from the chamber, at a rate of 1/48 min$^{-1}$, can be ignored. We again stress that $\gamma$ is not only the branching ratio for the initial VOC+OH reaction, but the fraction of reacted VOC molecules that become converted into HOM in the chamber, irrespective of detailed formation pathways. In steady state in JPAC, the concentration of HOM is constant, so

$$\frac{d[HOM]}{dt} = 0$$

and, therefore

$$k_1\gamma[VOC][OH] = k_{loss}[HOM]$$

Then, the molar HOM yield can be calculated as

$$\gamma = \frac{k_{loss}[HOM]}{k_1\gamma[VOC][OH]} \qquad (5)$$

For $k_{loss}$, we needed to assume that HOM condense irreversibly, which is a valid assumption based on earlier studies (e.g., Ehn et al., 2014). In accordance with Ehn et al. (2014), and verified in our experiments (not shown), kwall of 0.011 s$^{-1}$ was used. Average HOM concentrations for runs were calculated as a sum of all identified peaks with an oxygen content more or equal to six atoms. In case of phenol experiments, the peaks of the same composition were used as in benzene experiments for better comparison. The condensation sink was calculated using the following equation (Kulmala et al., 2012).

$$CS = 2\pi D \sum_{d_p} \beta_{m,d_p} d_{d_p} N_{d_p} \qquad (6)$$

where D is the diffusion coefficient for condensing vapour (determined by the molecular mass of each HOM) and $\beta_{m,\,dp}$ is the correction factor for the transition regime calculated based on the Fuchs-Sutugin approximation. $d_p$ is the diameter of particle size bins, and $N_{dp}$ is the concentration of particles in the chamber in the size bin $d_p$.

Finally, we stress that the HOM yield depends on our ability to determine the HOM concentrations, and is thus associated

with at least the same $\pm$ 70% uncertainty. Additional uncertainty will arise from the other parameters in Eq. (5), but these are likely to be much smaller than the uncertainty arising from HOM quantification. As stated earlier, only clearly identifiable peaks were utilized for HOM concentration calculations, in order to make the quantification as robust as possible. This means that we likely missed some HOM signal in the many smaller peaks that were unidentified or in the omitted peaks that showed contaminants at the same unit mass. Thus, our approach may cause an underestimation of the HOM yields.





## 3.5 Chamber kinetic model

In order to model HOM condensation during the seed addition in JPAC, we have constructed a simple kinetic model. The HOM mass concentration was modelled with 0.1 s resolution and the model assumed that the chamber was perfectly mixed for every time point. HOM molecular concentration for each point in time j for each HOM molecule i was calculated by adding

the HOM produced in a cm$^3$ in 0.1 s and subtracting the HOM lost from the HOM concentration in previous time point (j-1) as follows:

$$[HOM]_{j,i} = [HOM]_{j-1,i} + 0.1s \times (Production_{HOM} - Loss_{HOM})$$
$$= [HOM]_{j-1,i} + 0.1s \times (\gamma k_1[OH]_{j-1}[VOC]_{j-1} - k_{loss}[HOM]_{j-1,i}) \quad (7)$$

HOM molar yield $\gamma$ was set at 5% to reproduce the initial measured HOM concentration. [VOC] was set to constant 15.7

ppb (as measured by PTR-QMS), while [OH] concentration was scaled to match start and end measured HOM concentration (see Sect. 4.2.4). In Eq. (7), $k_{loss}$ took into account both wall loss and CS. The loss of HOM due to the flush out from the chamber was excluded as it is negligible compared to wall loss and CS in JPAC. Then, the total HOM mass concentration at point in time j equaled

$$[HOM\ mass]_i = \sum_{i=1}^{n} [HOM\ mass]_{j,i} = \sum_{i=1}^{n} \frac{[HOM]_{j-1,i} \times M_i}{N_A} \quad (8)$$

where $M_i$ is the molar mass of HOM molecule i and $N_A$ is Avogadro's constant.

## 4 Results and Discussion

### 4.1 Flow reactor study

In the first part of this work, we studied the OH oxidation of benzene, toluene and naphthalene in the Helsinki flow reactor using a nitrate-based CI-APi-TOF. In all three systems, we observed the formation of highly oxygenated organic molecules

(HOM). Product distributions are shown in Fig. 3 and include both HOM (products with six or more O-atoms) and less oxidized species, which were detected as adducts with $NO_3^-$. The following discussion focuses on peaks detected by adduct formation. We omit the reagent ion $NO_3^-$ when presenting the molecular formulas. However, the mass of the molecules refer to the correct mass, including the nitrate ion.

A few prominent peaks clearly dominated the benzene spectrum (Fig. 3a) with oxygen content of the products ranging

from 4 to 13 atoms. Among the closed-shell HOM, $C_5H_6O_7$, $C_5H_6O_8$, $C_6H_8O_8$, $C_6H_8O_9$, $C_6H_8O_{11}$ monomers and $C_{12}H_{14}O_8$, $C_{12}H_{14}O_{10}$, $C_{12}H_{14}O_{12}$, $C_{12}H_{14}O_{14}$ dimers dominated the signal. Also two radicals, $C_6H_7O_9\bullet$ and $C_6H_7O_{11}\bullet$, were detected.

The bicyclic peroxy radical (BPR), $C_6H_7O_5\bullet$ in the case of benzene, is potentially an intermediate in the formation of many HOM in the oxidation of aromatics. It was proposed in earlier studies that BPR from substituted aromatics can undergo further autoxidation (Molteni et al., 2018; Wang et al., 2017). In the case of benzene, it would form radicals with chemical composition

$C_6H_7O_x\bullet$, where x is an odd number larger than five. This hypothesis is supported by the presence of $C_6H_7O_9\bullet$ and $C_6H_7O_{11}\bullet$





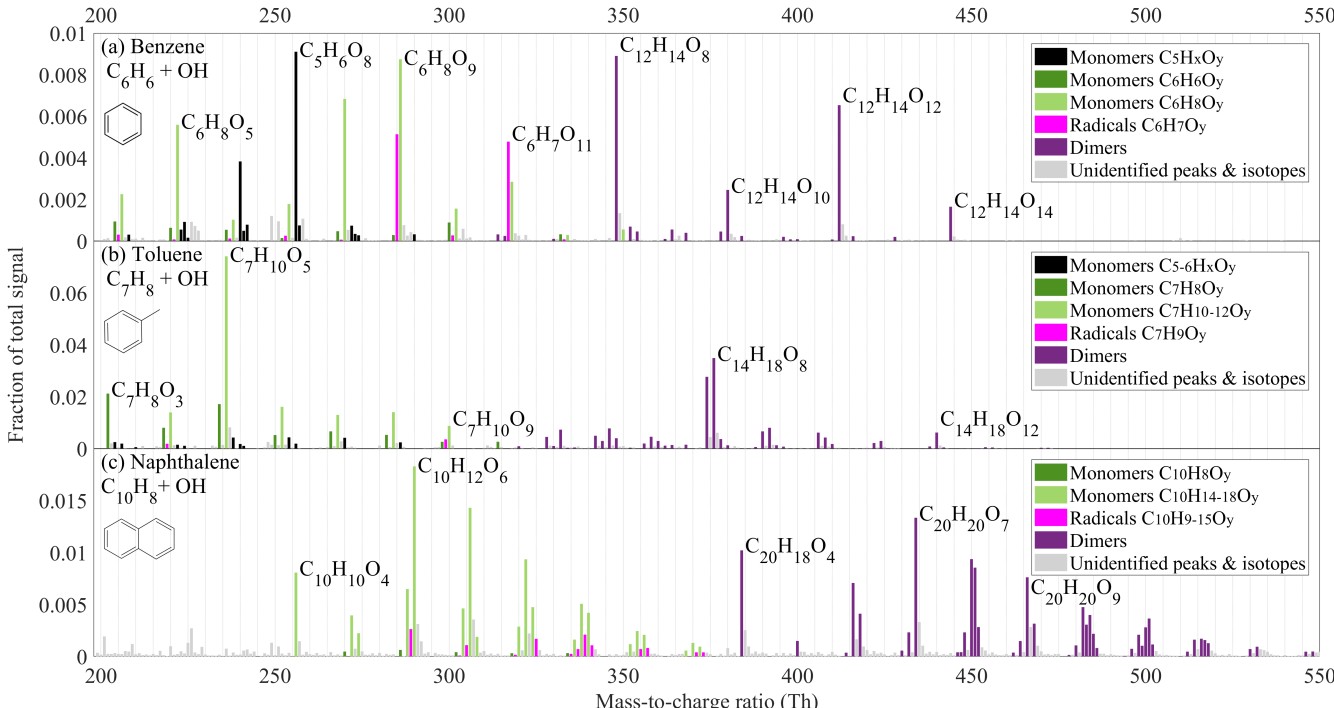

**Figure 3.** Spectra of organic oxidation products observed in oxidation of a) benzene, b) toluene and c) naphthalene. The y-axis shows the signal normalized by the total ion count of the instrument. The colours indicate different compound groups, as described in the legend of each subplot. The unit masses with more than one peak are marked with the colour of the most abundant peak. All the peaks above 200 Th are detected as adducts with $NO_3^-$, which is excluded from the labels. The full list of peaks can be found in Supplement (Tables S1-S3). Unidentified masses, isotopes and contaminant peaks are marked in light grey.

radicals, while BPR itself and the $C_6H_7O_7\bullet$ radical are detected only as very small signals. This is most likely due to the reduced detection efficiency for smaller radicals. While BPR is expected to have only one OH-group and its detection is unlikely, the reason for the low abundance of $C_6H_7O_7\bullet$ in the spectrum is unclear. It would be explained, however, if $C_6H_7O_7\bullet$ would contain two endoperoxides and a peroxy group, as also proposed by Molteni et al. (2018), therefore, still having only one OH group to

5  supply a hydrogen bond to a nitrate ion.

While $C_6H_7O_5\bullet$ (BPR) is weakly detected as such, we can observe products consistent with its termination reactions. For instance, $C_6H_8O_5$ at 222 Th can be formed through BPR reacting with $HO_2$. In photo-oxidation of any VOC, $HO_2$ production is efficient. $HO_2$ can also be produced in our reactor making a reaction with $HO_2$ an important bimolecular termination pathway in our system. This is also supported by the observation of $C_6H_8O_7$, $C_6H_8O_9$, $C_6H_8O_{11}$ and $C_6H_8O_{13}$.

10  The other important termination agents in our system are $RO_2$ radicals. The dominance of the $C_{12}H_{14}O_8$ dimer in the spectrum, likely formed from BPR self-reaction, strongly indicates the importance of ROOR' dimer formation. The prominence



of dimers with even oxygen numbers (also $C_{12}H_{14}O_{10}$, $C_{12}H_{14}O_{12}$, and $C_{12}H_{14}O_{14}$) is consistent with primarily odd-oxygen $RO_2$ being formed in the benzene system. If even-oxygen $RO_2$ were also abundant, odd-oxygen dimers, from cross reactions of odd- and even-oxygen $RO_2$, should be more prominent in the spectrum.

Monomer HOM with even number of O-atoms are also abundant in Fig. 3a, and these can be formed from $RO_2$ cross-reactions forming a carbonyl and an alcohol, or via alkoxy (RO) radical pathways. We are not able to separate formation pathways in such detail based on our data. However, the importance of RO radicals is suggested by some C5 radicals that we observed, namely $C_5H_7O_6\bullet$, $C_5H_7O_8\bullet$, and $C_5H_7O_9\bullet$, with one carbon less than benzene. Since benzene is a plain aromatic ring, loss of carbon from this molecule is only possible after a ring opening, potentially due to RO decomposition or another reaction causing the break of a bond between carbon atoms. After the ring is broken, CO or $CO_2$ could be lost and, after a reaction with another $O_2$, $C_5H_7O_x\bullet$ radicals are formed. These $RO_2$ radicals would terminate by reacting with $HO_2$ or another $RO_2$. Indeed, $C_{10}$ and $C_{11}$ dimers as well as closed-shell $C_5$ products are observed, of which $C_5H_6O_8$ (256 Th) is one of the dominant peaks in Fig. 3a. We cannot rule out other pathways for loss of carbon atoms from the molecules, and only conclude that it is a non-negligible pathway for HOM formation in benzene oxidation under our conditions.

The product spectrum from our flow reactor study of benzene oxidation shown in Fig. 3a was similar to the previous study by Molteni et al. (2018). For instance, the three largest signals in their study above 200 Th were $C_6H_8O_5$, $C_5H_6O_8$, and $C_{12}H_{14}O_8$, which are also prominent signals in our spectrum. Overall, almost the same molecules are present, with some variations in relative abundance. Specific differences worth noting are the larger fractions of $RO_2$ radicals visible in our spectrum, with two radicals ($C_6H_7O_9\bullet$ and $C_6H_7O_{11}\bullet$) being among the highest peaks. In contrast to our experiment, Molteni et al. (2018) observed dimers with odd and even amount of oxygen at comparable concentrations, suggesting the presence of both even- and odd-oxygen radicals in their system. While the specific experimental conditions between the studies of Molteni et al. (2018) and ours were not identical (benzene concentration ~100 times higher, residence time 50 % shorter, UV lamp irradiating part of the flow reactor in our study), some differences in the spectra are expected. However, based on the good agreement in product composition between the two studies, we conclude that direct photolysis of the VOC or its oxidation products (whether radicals or closed-shell species) were not affecting our results to a large extent. In addition, in our flow reactor, a direct photolysis experiment in absence of water showed no HOM formation. Nevertheless, future studies are needed to determine the exact role of photolysis in comparison to OH oxidation in initiating HOM formation in such systems.

In our flow reactor, we also tested the oxidation of toluene ($C_7H_8$, Fig. 3b). While the composition of toluene oxidation products generally is consistent with the reactions described for the benzene system above, some notable differences are observed. For instance, compared to the benzene experiment, the signal spreads out more evenly over many ions in the monomer product mass range, except for the dominant $C_7H_{10}O_5$ peak. Analogously to the benzene system, this is likely a termination product of a BPR in reaction with $HO_2$. Another difference is that even-oxygen dimers do not dominate the dimer spectrum. Instead, only two peaks, $C_{14}H_{18}O_8$ and $C_{14}H_{16}O_8$ are dominant. While the former dimer could originate from toluene-BPR self-reaction, the origin of the latter is unclear. We also observed some monomers with five or six carbon atoms, though at much lower contribution to total than the contribution of $C_5$ monomers in the benzene experiment. Overall, in comparison to the study by Molteni et al. (2018), where the toluene concentration was about 25 times smaller than in our experiments, many of the





peaks are similar. Specifically, it is interesting that in the toluene system, we also observed a few $C_7H_{12}O_{4-8}$ products with four hydrogen atoms more than in toluene itself, indicating the potential secondary OH oxidation (addition) step (Molteni et al., 2018). These products overlapped with $C_6H_8O_x$ compounds in the spectrum, so while they can be separated in high-resolution analysis, they are not recognisable in Fig. 3b.

In our naphthalene ($C_{10}H_8$) experiment, which is presented in Fig. 3c, the signal was distributed among an even larger amount of product peaks than in the toluene experiment. Interestingly, the largest monomer ($C_{10}H_{12}O_6$) contained 4 H-atoms more than the naphthalene precursor ($C_{10}H_8$). This suggests that an oxidation pathway including two OH attacks in combination with two $HO_2$-termination reactions was important in the naphthalene system. Evidence of $RO_2$ radicals formed through two OH attacks is also seen in $H_{20}$ dimers, which likely formed through cross-reaction of $H_9$ and $H_{11}$ $RO_2$. In the benzene spectrum we did not

observe any monomers containing four hydrogens more than the parent VOC, while in the toluene spectrum we observed only a minor fraction of such peaks. We attribute this to the higher VOC concentration used in our benzene experiment (400 ppm), in comparison to naphthalene (0.4 ppm) and toluene (25 ppm). The OH production from $H_2O$ photolysis stayed constant in our experiments, but the VOC acts as a sink for the OH radicals, which means that higher VOC concentrations will result in lower OH concentrations. This, in turn, decreases the likelihood of oxidation products reacting with OH a second time in our

flow reactor. Therefore, not only the competition between autoxidation and bimolecular $RO_2$ termination reactions will govern the exact concentration and distribution of HOM, but also the amount of secondary (or higher) OH attacks. For determining the importance of multi-generation OH oxidation as a source of HOM, longer time scales and lower VOC concentrations than reachable in our flow reactor were needed. Our further investigations in the JPAC chamber facility at the Research Centre Jülich were well suited for such a task.

**4.2    JPAC chamber studies**

**4.2.1    HOM yields**

To continue our study on the formation of HOM from aromatics, we performed systematic studies of benzene oxidation in the JPAC chamber (see Table 1). In JPAC, we were able to control the experimental conditions in more detail than in our flow reactor. In experiments without $NO_X$, the main parameters determining the oxidation process were the concentrations of OH

and VOC. As described in Sect. 3.1.2, these two parameters could be adjusted by changing the inflow of VOC or ozone, or by adjusting the photolysis rate by changing the gap width of the UV filter. Out of these, the input of VOC had the largest range, spanning around two orders of magnitude (1.6-112 ppb). An increase of VOC also meant a larger sink for OH, and thus the VOC and OH concentrations in the chamber were codependent.

     Figure 4a shows the measured HOM concentrations as a function of the VOC oxidation rate ($k_1 \times [VOC] \times [OH]$), including

primarily benzene experiments (square markers) but also three phenol experiments (circles). At small oxidation rates, the total HOM concentration increased linearly, but reached a plateau around 3-4 $\times 10^8$ cm$^{-3}$ at higher oxidation rate. If the loss coefficient ($k_{loss}$) of HOM were constant throughout all runs, experiments with the same HOM yield would fall on the same line. Assuming the loss of HOM is only determined by wall loss ($k_{loss} = k_{wall}$), the plotted lines in Fig. 4a would correspond to





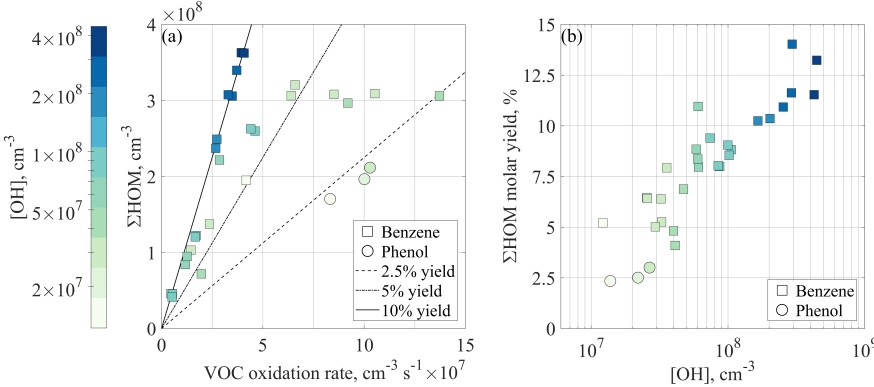

**Figure 4.** HOM concentrations and yields observed in the JPAC experiments. a) Total HOM concentration plotted as a function of VOC oxidation rate. If the HOM loss rate is constant between the experiments, conditions with the same molar HOM yields should fall on the same lines. The included lines (for 2.5, 5, and 10% yields, respectively) neglect the condensational sink (CS) and depict the yields in the case where loss to walls is the only sink for HOM. The color represents the concentration of hydroxyl radicals, squares depict benzene runs, while circles show phenol runs. b) Calculated HOM molar yields as a function of OH concentration in the chamber corrected for CS. Markers are the same as in panel a, as is the color code for easier comparison.

2.5, 5 and 10% yields. However, especially in the high [VOC] experiments (markers on the right hand side of Fig. 4a), the CS was of the same order as the wall loss and thus the approximation that $k_{loss}$ equals $k_{wall}$ is not valid anymore. In addition, the high-OH experiments (dark blue points) seem to result in the highest HOM yields.

    In order to identify the role of OH concentration for HOM yields, we calculated the molar yields, i.e. the number of HOM
molecules formed per reacted precursor VOC molecule, according to Eq. (5), properly accounting also for the CS. The results are shown in Fig. 4b. It is clear that under the conditions probed in JPAC, the main determining factor for the HOM yield was the OH concentration. The OH concentration in the chamber was clearly higher than in the atmosphere, but the average reaction time in the chamber was limited to approximately 48 minutes. If utilizing the concept of equivalent OH dose, a 48-minute residence time with $[OH] = 10^7 - 5 \times 10^8$ cm$^{-3}$ is equivalent to atmospheric oxidation times of roughly 10 h – 15 days
at OH concentration of $10^6$ cm$^{-3}$. In other words, our experiments span a reasonable range of atmospheric lifetimes.

    Our estimated HOM yields from benzene oxidation were 4.1-14.0%, which can be compared to a value of 0.2% provided by Molteni et al. (2018). Their value likely corresponds to the HOM yield of the first OH oxidation step, potentially also impacted by a second step, suggesting that more than 90% of the "HOM- forming potential" of benzene comes from multi-generation OH oxidation. In order to test the importance of the phenol pathway in HOM formation, three experiments were conducted
solely with phenol. Examining Fig. 4b, the phenol experiments show the lowest HOM yields (2.3 – 3%), suggesting that the phenol pathway is not the major route to form multi-generation HOM from benzene. However, the phenol experiments do not fall far from the trend produced by the benzene experiments, and thus phenol is likely to contribute to the total HOM formation from benzene.


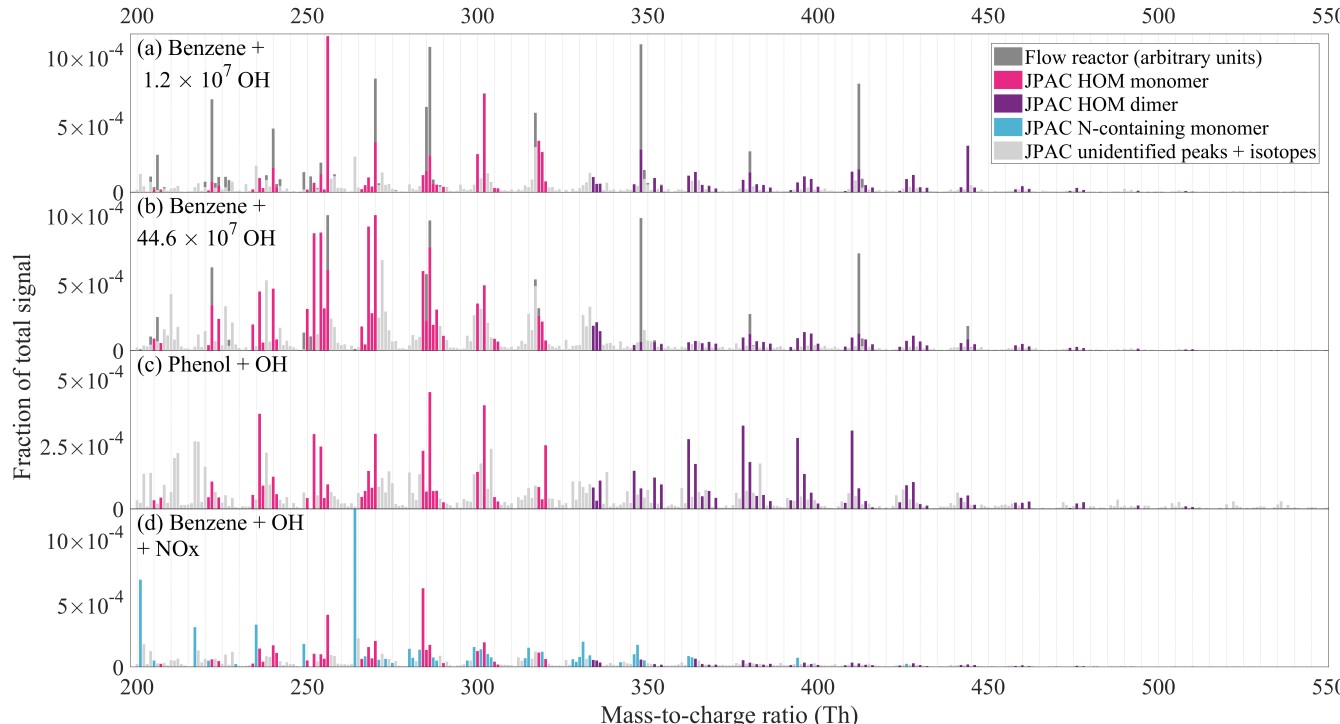

**Figure 5.** CI-APi-TOF spectra observed during experiments at JPAC. The y-axis shows the signal normalized to total ion counts. Panel a shows the mass spectrum of benzene oxidation at the lowest OH concentration among our experiments, while panel b corresponds to the highest OH concentration. In panel a and b, the scaled flow reactor mass spectrum is also included for comparison (dark grey bars, in arbitrary units). Panel c shows the oxidation products of a phenol oxidation experiment. In a, b and c, the colour schemes are identical and non-grey peaks represent those that were included in HOM yield calculation, with exception of few peaks. The full list of peaks is presented in Supplement (Table S4). Panel d shows the benzene + OH experiment in presence of $NO_X$, where N-containing ions ("N-containing monomers") dominate the spectrum.

### 4.2.2 Variability in HOM spectra

In addition to total HOM yield, OH concentration also affected the distribution of monomers and dimers in the benzene HOM spectrum, which is seen in Fig. 5 a-b. Higher OH concentration produced a spectrum with more peaks than did lower OH, indeed pointing at multiple oxidation steps. At lower OH, the monomers somewhat resembled the benzene flow reactor experiment. In Fig. 5a and b, the VOC oxidation rate is very similar ($4.2 \times 10^7$ versus $3.9 \times 10^7$ $cm^{-3}$ $s^{-1}$), while OH concentration is 35 times larger in Fig. 5b ($4.5 \times 10^8$ versus $1.2 \times 10^7$ $cm^{-3}$). The HOM yields corresponding to Fig. 5a and to Fig. 5b are 5.2% and 13.2%, respectively.

As OH increased in benzene experiments, an increase in the abundance of products with more H-atoms than the parent molecule due to secondary OH addition was expected; however, we observed an increase in products with lower H content, H





= 4-6. This means that OH oxidation through H-abstraction started to play a role. Oxidation of benzoquinone (formed in OH oxidation of phenol, MCM3.3.1 Bloss et al., 2005) could also potentially explain $H_{4-6}$ monomer HOM. After ring-opening, the BPR will contain one double bond, and if the products retain this, one more OH addition is possible. However, after this, OH oxidation can only proceed via H-abstraction, and if the subsequent termination reactions occur by loss of OH or $HO_2$, a

decrease in H-atoms will take place. In other words, it is to be expected, that multi-generation OH oxidation will produce also molecules with less H-atoms than the parent VOC.

The dimers detected in JPAC experiments had up to 18 oxygen atoms, which was larger than seen in flow reactor study. Dimers in JPAC had larger variability in the H-atom content, from 10 to 16. As in monomers, the dimer distribution also varied with OH concentration. At higher OH concentrations, a larger fraction of dimers was $C_{11}$ dimers suggesting more efficient

formation of $C_5$ radicals. At lower OH concentrations, the dimer distribution partly resembled the distribution seen in the flow reactor, while at higher OH, the dimer spectrum was distributed over a large number of peaks. In addition, in JPAC, the dimer-to-monomer ratio was observed to decrease with increased OH concentration. This may be explained by higher $HO_2$ concentrations at higher OH. Another possibility is that the dimer formation upon $RO_2$+$R'O_2$ reaction would be less likely for the $RO_2$ formed at high OH. The dimer formation rate has been shown to be highly dependent on the structure of the reacting

$RO_2$ (Berndt et al., 2018b).

In the phenol experiments (Fig. 5c), most elemental compositions were similar to those starting with benzene, as could be expected given that phenol has the same amount of C- and H-atoms as benzene. However, the relative distribution of peaks in the phenol spectrum did not directly resemble either the low or the high OH concentration benzene spectrum, again suggesting that a considerable fraction of HOM were produced from non-phenol pathways. In Fig. 5c, the peaks in colour

are the same peaks as were observed in the benzene experiments and were used for HOM yield calculations. Compared to the benzene experiments, phenol produced more dimers, of which $H_{12}$ dimers were a significant fraction, suggesting that $H_5$ radical production in phenol oxidation was somewhat more important than in benzene oxidation ($H_5$ and $H_7$ radicals would react to form $H_{12}$ dimers).

### 4.2.3   NO$_X$ experiment

While not being the main focus of this study, we also added $NO_X$ to the chamber in order to see its effect on HOM formation from benzene. Aromatic VOC and $NO_X$ are often co-emitted, and thus our no-$NO_X$ experiments are mainly relevant in places where the emissions were sufficiently diluted following transport from the vicinity of the sources. As seen in Fig. 5d, we observed many nitrogen-containing HOM as well as less oxidized compounds likely relating to nitrophenol-type compounds (i.e., nitrophenol with additional -OH or -$NO_2$ groups). The list of observed products is presented in Supplement (Table S4). In

addition, we observed also HOM without nitrogen, presumably from the reaction pathways involving alkoxy radicals (formed from $RO_2$+NO).

The nitrophenol-type compounds reacted much slower to changes in the chamber compared to HOM, likely indicating condensation and re-evaporation from chamber walls (i.e. semi-volatile compounds). As such, they can likely be transported long distances in the atmosphere, as shown in a recent study, which found a large nitrophenol signal in a CI-APi-TOF in the

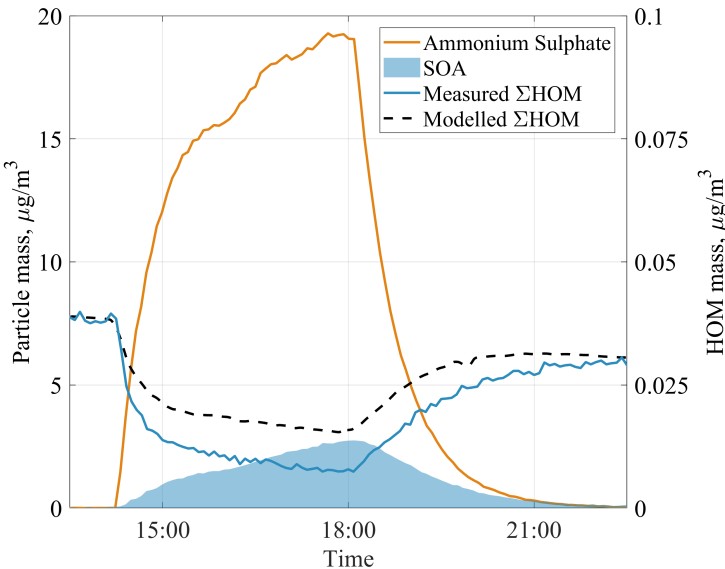

**Figure 6.** The evolution of components of aerosol mass and HOM mass concentrations during a seed aerosol experiment. The modelled total HOM concentration (dashed line) underestimates the removal of HOM from the chamber when aerosol seed is added, suggesting that not only HOM, but also some HOM precursors, decrease during the seed addition.

boreal forest (Yan et al., 2016). This study also showed that nitrophenol, despite only having one OH group, is readily detected by the CI-APi-TOF. Hyttinen et al. (2017) confirmed the stability of nitrophenol clusters with a nitrate ion using quantum chemical calculations.

### 4.2.4 HOM contribution to SOA

To examine the role of the observed HOM on SOA formation, we added ammonium sulfate seed aerosol to the chamber during one experiment. Aerosol addition increased the condensational sink for low-volatility species in the chamber. Many of these compounds would otherwise condense onto the chamber walls. Upon addition of the seed aerosol, HOM concentrations decreased and SOA concentrations increased (Fig. 6). Ehn et al. (2014) observed a similar behavior in JPAC experiments during ozonolysis of $\alpha$-pinene. In the Ehn et al. (2014) experiments, the condensed HOM explained more than 50% of the formed

SOA mass at 30 $\mu$g m$^{-3}$ total mass loading, including 7 $\mu$g m$^{-3}$ organics, while in our case the removed HOM explained around 30% at the highest total mass loading of 22 $\mu$g m$^{-3}$, which contained 2.7 $\mu$g m$^{-3}$ SOA mass. The chamber did not fully reach steady-state before the seed addition, which adds some uncertainty to the estimates, as discussed in the next paragraph. We expect that addition of seed aerosol will not affect the VOC oxidation rate, therefore VOC mass reacted should be the same over the whole experiment. Since the SOA yield is defined as SOA mass formed over VOC mass reacted, the observed increase

of SOA directly indicates an increased SOA yield. In our case, the calculated SOA yield was neglible before seed addition, and





increased to about 40% at the peak aerosol concentration of 22 $\mu$g m$^{-3}$. This clearly suggests that aerosol loadings can greatly influence SOA yield estimates from chamber studies as long as wall loss can compete with CS. We acknowledge that most SOA yield studies have been performed in larger chambers where the wall loss rate can be much smaller than in our chamber, and thus the effect is unlikely to be quite as large as observed here. The final SOA yield of 40% in our study is similar to 37%

yield in a low-NO$_X$ regime reported in previous work (Ng et al., 2007).

We constructed a simple chamber model to test the expected loss of HOM at different seed loadings, matching HOM concentrations to the periods before and after seed addition. HOM loss rates are a sum of wall loss rate (estimated as 0.011 s$^{-1}$) and condensational sink, which is calculated for every point in time according to Eq. (6), using the measured aerosol number size distribution. It should be noted that condensation sink assumes that the vapour is non-volatile. The reason behind

the HOM concentration not returning to the same level as before seed addition (~25% lower at the end of the experiment) is unclear. VOC in and outflow were stable, as were [O$_3$] and RH. As consequence, OH was also constant within the error ranges with a tendency to drop by about 10% over the time when seed aerosol was present. In our model we included a linear decrease of the OH concentration over the experiments to match the start and end HOM concentrations.

Using our model, we capture the shape of the HOM decrease very well, but find that our model underpredicts the loss of

HOM to the particles (Fig. 6). A possible explanation is that we underestimate the condensation sink (CS) or overestimate the wall loss rate (k$_{wall}$) in our model. For k$_{wall}$, the value would need to be ~2.5 times lower, corresponding to an inverse lifetime of 220 s, which is not supported by earlier experiments (Ehn et al., 2014) and observed lifetimes of individual HOM in our experiments. A similar correction factor of ~2.5 would be required for the CS in order to match the measurements, and this is a much greater value than the uncertainty in the aerosol loading data used for the CS calculation. In addition, the discrepancy

is larger for some of the detected HOM, while for others the model matches the observed loss (Fig. A1, A2 in Appendix A).

The most likely explanation for the mismatch in Fig. 6, Fig. A1, and Fig. A2, is that by introducing the seed aerosol we introduce a sink not only for the HOM detected by the CI-APi-TOF, but also for some precursors for multi-generation HOM formation that are undetected by our instrument (or detected at lower sensitivity). This explanation is plausible and is in support of our multi-generation HOM hypothesis. It also suggests that both the detected benzene-derived HOM and some of

the HOM precursors are sufficiently low-volatile to condense on 100 nm seed aerosol. If a HOM molecule were not to condense irreversibly onto the aerosol surface, it would lead to the opposite effect, i.e. that our model would overpredict the loss of HOM due to seed addition. Based on the explanation above, we note that our earlier estimate of HOM contribution to benzene SOA of 30% is a slight overestimation.

## 5   Summary and Conclusions

In this study, we confirmed the production of highly oxygenated organic molecules (HOM) in the OH-induced oxidation of aromatic compounds. We tested this chemical system in a flow reactor (10-second residence time) and in the Jülich Plant Atmosphere Chamber (JPAC; 48-minute residence time).





In benzene oxidation experiments in the flow reactor, we most likely observed first-generation HOM formed after a single OH attack. In experiments of toluene and naphthalene, we observed a broader distribution of HOM molecules, within which no particular compound clearly dominated the signal. We attributed this difference to lower VOC concentrations in the toluene and naphthalene systems compared to the benzene system, resulting in higher OH concentrations and consequent multiple OH reactions.

Complementary to the flow reactor study, we further investigated the multi-generation OH oxidation as a source for HOM in JPAC, specifically focusing on quantifying the benzene-derived HOM yield. The HOM molar yield, which in our definition included also multi-generation oxidation, in JPAC varied from 4.1 to 14.0% and strongly depended on the OH concentration. This dependence suggested that multi-generation oxidation produced a major portion of HOM. When examining the HOM composition, higher OH concentrations caused a larger variety in HOM products, with H-abstraction oxidation becoming more significant. We also noted a decrease in the dimer-to-monomer ratio as OH increased.

In a phenol oxidation experiment (a first-generation product of the benzene reaction with OH), we observed a lower HOM molar yield in comparison to the benzene oxidation at a comparable VOC oxidation rate and OH concentration. The lower HOM yield in phenol oxidation suggests that the non-phenolic pathway must be significant for HOM formation from benzene. This was further supported by the difference of the spectral distribution of HOM products between phenol and benzene.

Upon addition of about 4 ppb $NO_X$ to the benzene system in JPAC, we observed a production of N-containing HOM. These likely contained both nitrate- and nitro- functionalities. While termination reactions by $NO_X$ were significant, many HOM without nitrogen were still observed. The HOM spectrum observed in this experiment likely is more representative of the ambient urban air, where $NO_X$ concentrations are high. On the other hand, experiments without $NO_X$ are representative of the emissions after considerable dilution.

We also tested the ability of HOM from benzene oxidation to form secondary organic aerosol (SOA). We introduced seed aerosol to JPAC and investigated the rate of condensation of HOM. The loss of HOM was faster than the simple kinetic model predicted which likely means that also precursors for the detected HOM, which were not observed by our instrument, were condensing. This further supported our hypothesis that a large fraction of HOM in the benzene system was produced via multiple OH oxidation steps.

Our study confirmed the formation of HOM from aromatic compounds both on short and long time scales. We have determined the HOM yield from benzene oxidation at relevant atmospheric lifetimes. In addition, we examined the phenol branching pathway and confirmed the production of nitrogen-containing HOM upon $NO_X$ addition. Based on our findings, we conclude that HOM yield and composition is very sensitive to the reaction conditions. This sensitivity of HOM yield may partly explain the variability of SOA yield and time-dependency observed in previous studies. In addition, we conclude that atmospheric models should take into account HOM yield dependence on the chemical regime when implementing quantitative laboratory results. We also propose that future studies on aerosol formation from aromatic precursors would greatly benefit from including measurements of HOM in order to elucidate the detailed influence of experimental conditions on aromatic-derived highly oxygenated organic molecules and SOA, in the laboratory and the atmosphere.





*Data availability.* Data will be available from a persistent repository and upon request from corresponding authors

## Appendix A: Modelled condensation of individual HOM



**Figure A1.** Evolution of HOM monomers during the seed addition experiment.



**Figure A2.** Evolution of HOM dimers during the seed addition experiment.



*Author contributions.* ME, MPR, TM, JW, EK, IP designed the experiments. Instrument deployment and operation were carried out by IP, SS, OK, RT, TJB, MP, ÅMH, EK, JW and TM; data analysis was done by OG, TM, SS and RT; OG interpreted the compiled data set. OG, ME and MPR wrote the paper. All co-authors discussed the results and commented the manuscript. The authors declare that they have no conflict of interest.

5 *Competing interests.* OK works for Kärsa Oy and SS works for TSI GmbH.

*Acknowledgements.* This study was supported by the European Research Council (grant 638703, "COALA") and the Academy of Finland (grants 299574, 317380 and 320094), and Academy of Finland Centre of Excellence program (project 272041 and 307331). OG acknowledges Doctoral School in Atmospheric Sciences at the University of Helsinki (ATM-DP) for financial support. ÅMH acknowledges Formas (grant number 214-2013-1430) and Vinnova, Sweden's Innovation Agency (grant number 2013-03058), including support for her research
10 stay at Forschungszentrum Jülich. We also thank the tofTools team for providing tools for mass spectrometry data analysis.





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
