# Peer review of "Multi-generation OH oxidation as a source for highly oxygenated organic molecules from aromatics"

_Atmospheric Chemistry and Physics, 2019_

## Referee Comment (RC1) · Anonymous Referee #1 · 6 Aug 2019

General comments

Overall, this paper has shown that OH + aromatics (mainly benzene) produce a significant amount of highly oxidized molecules (HOM) that act as a reservoir for particles.

OH + VOC (aromatics) $\Rightarrow$ HOM $\Rightarrow$ particles

HOM are an array of compounds (with 6 or more O) that are the result of initial OH addition to the aromatics. Good evidence is provided that given enough time multiple OH reactions can take place and produce a larger array and amount of HOM.

The challenge for this type of work is how to parameterize the data. In the abstract,

they ask the modellers to consider these reactions, but this will only happen if such results are parameterized. At the moment this type of study is only semi-quantitative, i.e. there is an explanation for the HOMs but no rate coefficients. THIS IS THE BIG CHALLENGE.

Until a quantitative model is developed there is a problem of knowing which study is closest to atmospheric conditions. The study by Molteni(2018) indicated that HOM yields for benzene are 0.1-1% but in the present study, the yields are much higher 4 – 14%. Which is the correct result for atmospheric modelling? Are both studies in agreement, but it is the conditions that produce different yields?

Specific comments

page 7, 8 "Hydroxyl radicals were produced via photolysis of water at 184.9nm." I would make it clear that for each OH produce there is an HO2 (H + O2). This means in this reactor RO2 + HO2 is going to be significant and potentially suppress HOM formation. In the JPAC, OH is from O3/H2O (254nm) so only makes only OH

page 7, 10 "uncertainties in the VOC and OH concentrations were large enough that no quantitative analysis was attempted" Below an estimate [VOC] is made, so can an estimate of OH be made? What is [H2O] approximately?

page 8, 12 "The UV lamp (Philips, TUV 40W, $\lambda$max = 254nm) was located inside the chamber and was shielded from both ends with UV-absorbing glass tubes." How even is this light distributed in the chamber?

From Figure 2, it takes 5 hours before reaching steady-state. Is this time to SS representative of all the experiments? If I understand correctly, you are waiting for

OH + VOC (aromatics) $\Rightarrow$ HOM $\Rightarrow$ particles

to reach steady-state and the MS data is only analyzed once SS is reached? Is there valuable data in the first 5 hours? If the HOM ratios are changing during this build-up time there should be further clues to the HOM formation mechanism? In the Helsinki

experiments you have not reached SS, so where along this HOM concentration curve is applicable? DOES ANYONE HAVE A KINETIC MODEL TO EXPLAIN FIGURE 2?

In Equation 5, $\gamma$ is in the denominator.

page 15, 4 "for each HOM molecule i" Can you indicate the value of i, i.e. how many HOMs are considered?

page 16, 8 "HO2 can also be produced in our reactor making a reaction with HO2 an important bimolecular termination pathway in our system." Please make in clear in the experimental that HO2 is made from H2O photolysis.

Is there any specific reason that 185 nm was used to generate OH. If 254 nm and O3/H2O was used then there would be a more direct comparison to the JPAC experiment. Can the lamp be translated along the reactor to change the contact time? I note that Molteni(2018) used an excimer lamp at 172 nm. Both 185nm and 172nm will photolyze aromatics, but this is not the case for 254 nm.

The results in Figure 3 show that a range of HOMs are formed and the text provides some explanation of the type of reactions required to make each HOM. I was wondering if this could be quantified further. For instance, the dimers require an RO2 + RO2 bimolecular reaction. What sort of rate coefficient is required? While you state you do not know OH concentrations in the Helsinki experiments this makes it a difficult question, but in the JPAC experiments maybe this is possible.

page 18, 13 "The OH production from H2O photolysis stayed constant in our experiments, but the VOC acts as a sink for the OH radicals, which means that higher VOC concentrations will result in lower OH concentrations." Is there a reason why benzene or toluene was not lowered to maybe promote 2 OH reactions? If you had the bubbler before the MFC would you have greater control of the [VOC]?

In the Helsinki experiments, if the residence time could be changed then the mechanism of how these HOMs are formed might be clearer and even provide some kinetic

assignments. If the lamp was located further along the reactor would this significantly change the residence time?

Page 18, 31 "At small oxidation rates, the total HOM concentration increased linearly, but reached a plateau around 3-4 $\times$ 108 cm-3 at higher oxidation rate." This plateau is true for the light green square but not for the dark blue!

From Figure 4 b, HOM yield appears to be linear in OH. Surely there must be a limit where the HOM yield plateaus? If the HOM yield is increasing with more multiple OH reactions, does this mean that HOMmulti $\Rightarrow$ particles is slower?

Page 19, 11 "Our estimated HOM yields from benzene oxidation were 4.1-14.0%, which can be compared to a value of 0.2% provided by Molteni et al. (2018). Their value likely corresponds to the HOM yield of the first OH oxidation step, potentially also impacted by a second step, suggesting that more than 90% of the "HOM- forming potential" of benzene comes from multi-generation OH oxidation." I'm confused here as to which study is relevant to the atmosphere. Molteni(2018) had a factor of ten less HOM than the present study, is this consistent with the OH concentration they used in their experiment? Which HOM yield would go into a model? Can you provide more discussion in the paper about this difference? Have you artificially raised the [OH] in order to bring about large [HOM]?

If you looked at the HOMi signal at early times, the 5 hours before steady-state, can you see the various HOMi evolving in time?

Now if you are saying that the HOM yield is 10 times smaller for 1 OH reaction, then should the HOM spectra dramatically change, i.e. big difference between Figure 3 and 5?

Page 21, 1 "However, after this, OH oxidation can only proceed via H-abstraction, and if the subsequent termination reactions occur by loss of OH or HO2, a 5 decrease in H-atoms will take place. In other words, it is to be expected, that multi-generation OH

oxidation will produce also molecules with fewer H-atoms than the parent VOC." Are these high [OH] HOM relevant to the atmosphere?

Page 21, 3 "Another possibility is that the dimer formation upon RO2+R'O2 reaction would be less likely for the RO2 formed at high OH." I would have thought RO2 + RO2 is more likely the higher the OH concentration.

Page 23, 22 "but also for some precursors for multi-generation HOM formation that are undetected by our instrument (or detected at lower sensitivity)." But I thought you are saying that the multi-generation HOM are from the primary generation HOM.

OH + HOMprimary $\Rightarrow$ HOMmulti

What sort of intermediate before HOMmulti do you think might be present that is not detected? I understand why BPR is not detected but not what you are presently suggesting.

Page 24, 30 "In addition, we conclude that atmospheric models should take into account HOM yield dependence on the chemical regime when implementing quantitative laboratory results." Can you make it clear which study is relevant to the atmosphere, the present work (high HOM yield) or Molteni(2018) (low HOM yield).

---

## Referee Comment (RC2) · Anonymous Referee #2 · 27 Aug 2019

General Comments:

The authors have reported on a series of experiments aimed at quantifying the total and speciated HOM formed from the oxidation of important aromatic VOCs using two different facilities: a flowtube and an environmental chamber. The data analysis is enlightening and a number of useful interpretations are made that will help the field continue to advance understanding of this potentially impactful atmospheric phenomenon. In general though, I found the lack of quantitative conclusions disappointing. The yield estimates provided are somewhat provocative because they are so high compared to previous reported values, but it is unclear how the atmospheric model

community should use these estimates, or if they should use them at all. I recommend this manuscript be published, but I have some reservations with the overall messaging.

The bottom lines appear to be, 1) HOMs from aromatics are confirmed and 2) when constructing an laboratory-based model, everything matters. If the authors could extend their message to how their findings relate to HOM formation from biogenics (both in terms of magnitude and sensitivity to chamber conditions), which is comparatively well-studied and which has been implemented in some large scale models, it might enhance the narrative. A key point I was missing was the authors' interpretation of how important these processes are in and downwind of the urban atmosphere. Do they have some idea if aromatic oxidation can be a substantial source of ultrafine particle formation events, or is it still too early to tell?

Specific Comments

1. Definition of molar yield: I appreciate the authors decision to use an operational definition of HOM yield, which is more in line with traditional methodologies for estimating SOA formation in large-scale models. In the short term, this option is easier to transcribe into models. However, the cost is that it is harder to account for sensitivity to environmental conditions like OH and NOx and translate chamber residence time to variable model time steps. Clearly, this approach of lumping the multigenerational formation together with the prompt formation is more suitable for aromatic VOCs (a point the authors make), but the timescale for this formation given on Page 19, line 9-10 is 10 h – 15 days. Some global models run with chemistry time steps on the order of tens of minutes to multiple hours. But regional models can be in the range of 5 minutes down to 45 seconds for high resolution cases. And LES models are even faster. Can the authors please consider discussing this aspect of how their data will be used? Are yields defined in this way really useful to large-scale models or are they more useful to other experimental efforts trying to constrain the total HOM formed.

2. The authors' point that OH and NOx should be considered when predicting HOMs

from aromatics is well-taken; they have demonstrated it well. However, can they provide a parameterization for these effects that can be explored by other groups. The only option they have left the reader with is to interpolate the data in Table 1. If the authors do not think such a parametrization would be helpful, please discuss why.

3. Section 3.5: I recommend adding a table of all of the HOM molecules considered for the kinetic model of the seed experiment, to the appendix or supplemental. Are ions reported because multiple formulae apply to one ion in some cases? Equations 6, 7, and 8 indicate the kinetic model needs molecular weight and diffusion coefficent in air of every species of interest. If so, these parameters could be reported as well. I also recommend adding the formula to each of the sub-panels in Figures A1 and A2.

4. Page 15, line 9: HOM molar yield is set to 5%? Apparently I am confused about this model. I was under the impression that each HOM molecule would have a specific yield based on its relative abundance in the spectra from the base experiment. Is this not the case? Please consider explaining this portion of the approach clearer in the text.

5. The authors connect their experiments to previous SOA studies to try to explain the variability seen in the literature with what they have learned about HOMs. But the only parameter discussed is seed aerosol concentration. Are there any other features of HOM formation the authors think are connected to the apparent variability in historical SOA yields?

Typos/Suggestions

1. Table 1 caption: quadrupole?

2. Page 13, Line 6-7: Are you reporting SOA yields in this work?

3. Page 14, Line 5-6: Please provide an equation relating k_loss to k_wall and CS.

4. Page 14, Line 27-29: What is the magnitude of uncertainty introduced from unidentified or omitted peaks?

[Figure]

5. Page 22, Line 9-11: Please rewrite this sentence to make it clearer that "total mass loading" is total PM mass (organic + inorganic) and that the percentage numbers are relevant to the organic only mass. There's nothing incorrect about this as-is sentence, but it could be reordered to make it easier for the reader to digest quickly.

6. Page 23, line 2-3: This statement "This clearly suggests. . .compete with CS." has been shown in other studies documenting the effect of seed aerosol and they should be referenced.

7. Figures A1 and A2: Do the authors have an explanation for the range of variability observed for different m/z. Some of them are relatively smooth, and others change wildly. Are the latter intermediates?

---

## Author Comment (AC1) · 31 Oct 2019

We thank the reviewers for their comments. Below, we present responses, labeled as Rn (n being the number of the response) for easier referencing, to each comment in blue colour.

**Response to Anonymous Referee #2**

General Comments:

The authors have reported on a series of experiments aimed at quantifying the total and speciated HOM formed from the oxidation of important aromatic VOCs using two different facilities: a flowtube and an environmental chamber. The data analysis is enlightening and a number of useful interpretations are made that will help the field continue to advance understanding of this potentially impactful atmospheric phenomenon. In general though, I found the lack of quantitative conclusions disappointing. The yield estimates provided are somewhat provocative because they are so high compared to previous reported values, but it is unclear how the atmospheric model community should use these estimates, or if they should use them at all. I recommend this manuscript be published, but I have some reservations with the overall messaging.

R1. Both anonymous reviewers made us aware that our statements (in abstract and conclusions) about model implementations of SOA and HOM were unclear and somewhat misleading. We therefore will clarify our intentions here, as they are relevant for several later comments as well.

In this manuscript, we did not aim at parameterisations for atmospheric models. Nor would we have been able to, as only a limited set of atmospheric conditions and oxidation products were sampled in our experiments. Instead, we tried to point out that the yields of both SOA and HOM are likely to be extremely sensitive to variations in the specific conditions, and thus modelers need to carefully consider which atmospheric conditions a certain laboratory study result is applicable. In this work, we show that the OH concentration affects both total HOM yield as well as HOM composition. OH has been a less frequently considered parameter in HOM research so far, but of course is critical in aromatic oxidation. It is clear that more studies are needed to understand complex and coupled processes of aromatic oxidation to be able to properly model it. We think that our quantitative results increase the general understanding of HOM formation from aromatic systems; however, the absolute values are most useful for other experimentalists to interpret and plan their HOM and SOA studies. We have clarified the relevant sentences in the *Abstract* and *Summary and Conclusions* as follows:

Abstract: "Based on our results, we conclude that HOM yield and composition in aromatic systems strongly depend on OH and VOC concentration and more studies are needed to fully understand this effect in formation of HOM and, consequently, SOA."

Conclusions: "In addition, we conclude that more studies are required to fully understand how HOM yield and composition in aromatic systems depends on OH concentration and how the differences in HOM will affect the rate and magnitude of SOA formation. It would be valuable to sample different time scales, low and high reactant concentrations as well as effect of other important parameters, such as the effect of lights and NOx"

The bottom lines appear to be, 1) HOMs from aromatics are confirmed and 2) when constructing an laboratory-based model, everything matters. If the authors could extend their message to how their findings relate to HOM formation from biogenics (both in terms of magnitude and sensitivity to chamber conditions), which is comparatively well-studied and which has been implemented in some large scale models, it might enhance the narrative.

R2. We agreed with the reviewer and added a paragraph about the comparison to biogenic HOM studies in the end of section 4.2.1.:

"In comparison to biogenic VOC, our results were closest to the HOM yields observed in ozonolysis of α-pinene and limonene, 3.4-7% and 17% respectively (Bianchi et al. 2019). In the biogenic systems, especially if a VOC contains an endocyclic double bond, the first oxidation step by $O_3$ is known to form HOM at large yields. On the other hand, the observed yields in first-step OH oxidation are reported to be low (~1%, Bianchi et al. 2019). To our knowledge, no studies exist that explore HOM yields of biogenic VOC oxidation as a function of OH concentration. However, McFiggans et al. (2019) indicated a non-linear increase of HOM concentration with increasing α-pinene oxidation rate. We would therefore expect that also in biogenic systems, an increase in HOM yield due to multi-generation OH oxidation could be observed."

A key point I was missing was the authors' interpretation of how important these processes are in and downwind of the urban atmosphere. Do they have some idea if aromatic oxidation can be a substantial source of ultrafine particle formation events, or is it still too early to tell?

R3. Based on our laboratory results alone, it is not possible to tell what will be the contribution of these HOM to ambient particle formation, i.e. cluster formation. However, based on previous studies of HOM as well as our SOA experiment results, it is reasonable to think that these HOM are low-volatility compounds, which in the ambient atmosphere will be able to condense onto small aerosol particles. Clearly, as we also see HOM production in presence of NOx and we know other aromatics also produce HOM (Molteni et al. 2018, Wang et al. 2017, Hammes et al. in review), these compounds will be important players in particle growth and SOA formation where aromatic VOCs are abundant.

We added to *Summary and Conclusions*:

"Based on current understanding of HOM as well as our SOA experiment result, we can suggest that HOM observed in this study may play an important role in initial particle growth in ambient atmosphere where aromatic VOCs are abundant."

Specific Comments

1. Definition of molar yield: I appreciate the authors decision to use an operational definition of HOM yield, which is more in line with traditional methodologies for estimating SOA formation in large-scale models. In the short term, this option is easier to transcribe into models. However, the cost is that it is harder to account for sensitivity to environmental conditions like OH and NOx and translate chamber residence time to variable model time steps. Clearly, this approach of lumping the multigenerational formation together with the prompt formation is more suitable for aromatic VOCs (a point the authors make), but the timescale for this formation given on Page 19, line 9-10 is 10 h – 15 days. Some global models run with chemistry time steps on the order of tens of minutes to multiple hours. But regional models can be in the range of 5 minutes down to 45 seconds for high resolution cases. And LES models are even faster. Can the authors please consider discussing this aspect of how their data will be used? Are yields defined in this way really useful to large-scale models or are they more useful to other experimental efforts trying to constrain the total HOM formed.

R4. We would like to mention that our data set is too limited and the reviewer's requests for generalizations are beyond the scope of this experimental study. The absolute HOM yield values are mainly useful for other experimentalists to plan and interpret their studies, while the overall message that HOM yield and composition depends on OH is useful for experimentalists and modelers alike. Please see also the response R1.

2. The authors' point that OH and NOx should be considered when predicting HOMs from aromatics is well-taken; they have demonstrated it well. However, can they provide a parameterization for these effects that can be explored by other groups. The only option they have left the reader with is to interpolate the data in Table 1. If the authors do not think such a parametrization would be helpful, please discuss why.

R5. Please see responses R1 and R4 above. We have performed only one experiment with $NO_X$, which was good to demonstrate that $NO_X$ should be considered, but not sufficient to formulate parametrizations. A recent paper by Hammes et al. (in review), discusses $NO_X$ dependence of HOM formation in TMB in more detail. For yield dependence on OH/VOC/JO1D, all data is provided in Table 1, which can be used by other groups for comparison. These results are valid for a certain range of studied concentrations and the system appears that simple parametrizations are not so easy: the reaction system is too complex.

In addition, we added a short clarification to the discussion in section 4.2.1.:

"It should be noted that the specific dependency of HOM yield on OH may vary if other gases and loss mechanisms would be present."

3. Section 3.5: I recommend adding a table of all of the HOM molecules considered for the kinetic model of the seed experiment, to the appendix or supplemental. Are ions reported because multiple formulae apply to one ion in some cases? Equations 6, 7, and 8 indicate the kinetic model needs molecular weight and diffusion coefficient in air of every species of interest. If so, these parameters could be reported as well. I also recommend adding the formula to each of the sub-panels in Figures A1 and A2.

R6. The peaks used for the kinetic model in seed experiment (Figures A1 and A2) are identical to the ions used for HOM yield calculations. The list of them is presented in Table S4 in Supplement under "Non-nitrogen containing HOM". We identified the composition of ions using high resolution peak fitting and chose the peaks that were mainly single peaks for calculations, meaning other ions had minor contribution to the peak at this mass (see section 3.2.1 of the main text). We then performed unit mass resolution analysis for calculating HOM concentration (meaning we integrated the signal over the chosen peaks), and therefore we chose to show values for m/z in Figures A1 and A2 instead of ion composition. The ion composition at a specific mass can be found from Table S4.

In Equations 7 and 8, $M_i$ was determined as the mass of the observed peak minus the mass of $NO_3^-$, if the molecule was clustered with $NO_3^-$.

We have made few modifications to make these clearer:

1. In Methods:

"If a HOM molecule was detected as cluster with $NO_3^-$, $M_i$ was calculated as the m/z value of the peak minus the mass of $NO_3^-$. For this model, we used the peaks corresponding to the same HOM molecules as in the

HOM yield calculation, 69 peaks in total. The list of the m/z values and corresponding compositions can be found in Table S4 in Supplement."

2. In caption of Figure A1/A2

"Figure A1. Evolution of measured and modelled HOM monomers during the seed addition experiment. The list of HOM compositions for each peak at corresponding m/z is presented in Table S4 in Supplement."

"Figure A2. Evolution of measured and modelled HOM dimers during the seed addition experiment. The list of HOM compositions for each peak at corresponding m/z is presented in Table S4 in Supplement."

3. In caption of Table S4 in Supplement.

"Table S4. Peaks identified in JPAC. Non-nitrogen containing HOM were included in HOM yield calculation and kinetic model for seeded experiment…"

Indeed, in Equation 6, a diffusion coefficient is needed. Diffusion coefficient was approximated as 0.06 $cm^2$/s using an approximate value for weighted mean molecular weight 237 g/mol and diffusion volume as 170 based on method described by Fuller et al. 1966.

We corrected the text in methods to clarify this:

"D was approximated as 0.06 $cm^2$/s based on the mean molar mass 237 g/mol and approximated diffusion volume 170 of the observed HOM, according to the approach described by Fuller et al. 1966."

4. Page 15, line 9: HOM molar yield is set to 5%? Apparently I am confused about this model. I was under the impression that each HOM molecule would have a specific yield based on its relative abundance in the spectra from the base experiment. Is this not the case? Please consider explaining this portion of the approach clearer in the text.

R7. The reviewer's impression was correct, and we have modified the sentence in Methods to clarify that the "HOM molar yield" referred to the sum of all detected HOM molecules in the spectra:

"The molar yield of total HOM was set to 5% to match the measured HOM concentration before seed addition. For an individual $HOM_i$, the relative abundance in the spectra determined its yield."

5. The authors connect their experiments to previous SOA studies to try to explain the variability seen in the literature with what they have learned about HOMs. But the only parameter discussed is seed aerosol concentration. Are there any other features of HOM formation the authors think are connected to the apparent variability in historical SOA yields?

R8. We observe increase in HOM molar yield with an increasing OH. At the same time, we see that HOM readily condenses onto the seed aerosol and contributes to SOA formation. This means that variability in HOM will affect the formation of SOA. With our results, we could for example explain why at lower VOC concentrations in toluene oxidation Chen et al. (2019) observed larger SOA yields. At lower VOC concentration, there was more OH available per molecule of VOC likely allowing for HOM formation due to multi-generation OH oxidation. They performed SOA experiments without seed addition, so low-volatility compounds were needed to drive the growth of small particles.

Drawing more general conclusions is hard as many SOA studies do not investigate OH effect in detail. In addition, some studies lack wall loss corrections. Therefore, we cannot readily decouple many parameters affecting SOA yields reported in the literature. In context of SOA studies, we expect that HOM influence will be most visible for unseeded SOA formation as well as when seed is injected at low concentrations (Ehn et al. 2014). Therefore, in those experiments, we would expect SOA to depend on OH as well.

We added to the end of section 4.2.4 an explanation how we think our result in HOM will affect SOA:

"Based on our current understanding of HOM and the results from our SOA experiment, we expect that the change of HOM yield with OH would affect in turn the formed SOA yield. It is likely, that this effect will be mainly pronounced in SOA studies conducted without seed aerosol or in studies where seed aerosol is added at low concentrations (Ehn et al. 2014)."

Typos/Suggestions
1. Table 1 caption: quadrupole?
R9. We fixed this.

2. Page 13, Line 6-7: Are you reporting SOA yields in this work?
R10. We report SOA yield of 40% during the seeded experiment in section 4.2.4.

3. Page 14, Line 5-6: Please provide an equation relating k_loss to k_wall and CS.
R11. We added the equation to the text: k_loss = k_wall + CS.

4. Page 14, Line 27-29: What is the magnitude of uncertainty introduced from unidentified or omitted
    peaks?
R12. In Figure 5a and b, peaks included into yield calculation constitute ~50% of the signal between masses 200 and 550. This means that the yield can be potentially underestimated by a half. We included this into the text:

"As stated earlier, only clearly identifiable peaks were utilized for HOM concentration calculations, in order to make the quantification as robust as possible. These peaks constituted approximately 50% of the total signal in mass range from m/z 200 to 550. Although isotopes account for some of this unexplained fraction, our approach may cause an underestimation of the HOM yields by up to 50%."

5. Page 22, Line 9-11: Please rewrite this sentence to make it clearer that "total mass loading" is total PM
    mass (organic + inorganic) and that the percentage numbers are relevant to the organic only mass.
    There's nothing incorrect about this as-is sentence, but it could be reordered to make it easier for the
    reader to digest quickly.

R13. We re-wrote the sentence as follows:

"In the Ehn et al. (2014) experiments, the total aerosol mass loading was 30 $\mu g\ m^{-3}$, out of which 7 $\mu g\ m^{-3}$ was the SOA mass formed during the experiment. The condensed HOM explained more than 50% of that SOA mass. In our benzene case, at total aerosol mass loading of 22 $\mu g\ m^{-3}$, the removed HOM explained around 30% of the 2.7 $\mu g\ m^{-3}$ SOA mass."

Page 23, line 2-3: This statement "This clearly suggests. . .compete with CS." has been shown in other studies documenting the effect of seed aerosol and they should be referenced.

R14. We have added references as follows:

"This clearly suggests that aerosol loadings can greatly influence SOA yield estimates from chamber studies as long as wall loss can compete with CS (Ehn et al. 2014, Kokkola et al. 2014, Zhang et al. 2014)."

6. Figures A1 and A2: Do the authors have an explanation for the range of variability observed for different m/z. Some of them are relatively smooth, and others change wildly. Are the latter intermediates?

R15. The effect is simply due to differences in the absolute abundance of the different m/z, with low-signal m/z appearing more noisy.

**Response to Anonymous Referee #1**

General comments

Overall, this paper has shown that OH + aromatics (mainly benzene) produce a significant amount of highly oxidized molecules (HOM) that act as a reservoir for particles. OH + VOC (aromatics) ⇒ HOM ⇒ particles HOM are an array of compounds (with 6 or more O) that are the result of initial OH addition to the aromatics. Good evidence is provided that given enough time multiple OH reactions can take place and produce a larger array and amount of HOM. The challenge for this type of work is how to parameterize the data. In the abstract, they ask the modellers to consider these reactions, but this will only happen if such results are parameterized. At the moment this type of study is only semi-quantitative, i.e. there is an explanation for the HOMs but no rate coefficients. THIS IS THE BIG CHALLENGE.

R16. Please refer to the response R1.

Until a quantitative model is developed there is a problem of knowing which study is closest to atmospheric conditions. The study by Molteni(2018) indicated that HOM yields for benzene are 0.1-1% but in the present study, the yields are much higher 4 – 14%. Which is the correct result for atmospheric modelling? Are both studies in agreement, but it is the conditions that produce different yields?

R17. Molteni et al. (2018) provided a value of 0.2% for HOM molar yield in oxidation of benzene. Their experiment is conducted at shorter residence time and corresponds to mainly first and some second OH oxidation steps, as is supported by their HOM composition. Our flow reactor experiments are closer to the set-up of Molteni et al. and so is the HOM product distribution. We could not quantify our flow reactor results, however. Our results for HOM yield in JPAC chamber, on the other hand, correspond to HOM formed over long residence time, allowing for sequential OH oxidation steps, and also for slower isomerization reactions propagating the oxidation sequence. Therefore, the results are not in contradiction, but are relevant for different regimes. To provide parametrisations for atmospheric models, more studies on both long and short time-scales at atmospherically relevant concentrations are needed.

We modified and re-arranged the text in section 4.2.1 where the yields are compared, also including the issue with different timescales:

"Our estimated HOM yields from benzene oxidation were 4.1-14.0%, which can be compared to a value of 0.2% provided by Molteni et al. (2018). The difference in the results is expected due to the substantial

difference in the studied timescales (20 second in their study). In addition, in their flow reactor, air parcel was exposed to an initial OH concentration that decreased as OH reacted away, while in JPAC, the OH was produced continuously. This resulted in different OH doses in the systems. Considering these differences, less oxidation steps would be expected in a flow reactor. As a result, the yield in Molteni et al. (2019) likely corresponds to the HOM yield of the first OH oxidation step, potentially also impacted by a second step. This suggests that more than 90% of the "HOM- forming potential" of benzene comes from multi-generation OH oxidation in combination with slower isomerization reactions that may not be observed on shorter time scales."

We also added short clarifications into *Methods* and *Results and Discussion* about an OH dose:

"The OH concentration integrated over the residence time defines an OH dose, which could be used to compare the results with other systems or atmosphere. By definition, the OH dose would recognise that a 48-minute experiment with OH concentration of $10^8$ cm$^{-3}$ is equivalent to a 480-minute experiment with OH concentration of $10^7$ cm$^{-3}$. Since in our JPAC experiments the residence time is kept constant, we use the OH concentration to describe our system."

**Specific comments**

page 7, 8 "Hydroxyl radicals were produced via photolysis of water at 184.9nm." I would make it clear that for each OH produce there is an HO2 (H + O2). This means in this reactor RO2 + HO2 is going to be significant and potentially suppress HOM formation. In the JPAC, OH is from O3/H2O (254nm) so only makes only OH

R18. We agree that in our flow reactor HO$_2$ formation is considerable. With each OH, we co-produce HO$_2$. On the other hand, each OH will also produce RO$_2$ through VOC oxidation. We can see that RO$_2$+RO$_2$ reactions are also important in our reactor as we can see HOM dimers. In the flow reactor, we still see HOM with 13 oxygen atoms, suggesting that autoxidation was fast enough to compete with bimolecular termination reactions.

We modified the text in the section 4.1 to reflect this:

"In photo-oxidation of any VOC, HO$_2$ production from RO$_2$ is efficient. HO$_2$ will also be co-produced from water photolysis in our reactor making a reaction with HO$_2$ an important bimolecular termination pathway. This is supported by the observation of $C_6H_8O_7$, $C_6H_8O_9$, $C_6H_8O_{11}$ and $C_6H_8O_{13}$. However, high oxygen content of HOM as well as the existence of dimeric species shows that the termination of RO2 by HO2 was not a dominant process in our system."

page 7, 10 "uncertainties in the VOC and OH concentrations were large enough that no quantitative analysis was attempted" Below an estimate [VOC] is made, so can an estimate of OH be made? What is [H2O] approximately?

R19. Unfortunately, we cannot estimate the OH concentration as the intensity of the lamp is unknown and thus the water photolysis rate cannot be determined.

page 8, 12 "The UV lamp (Philips, TUV 40W, λmax = 254nm) was located inside the chamber and was shielded from both ends with UV-absorbing glass tubes." How even is this light distributed in the chamber?

R20. The light distribution in the JPAC chamber is not completely even. Direct measurement of OH by LIF showed that the OH concentrations at the further end of JPAC were ½ of the average OH concentration determined by VOC consumption measurements (method applied here). However, the chamber is constantly stirred (mixing time < 2min) which ensures that the reactants see on average the same OH field.

From Figure 2, it takes 5 hours before reaching steady-state. Is this time to SS representative of all the experiments? If I understand correctly, you are waiting for

OH + VOC (aromatics) $\Rightarrow$ HOM $\Rightarrow$ particles

to reach steady-state and the MS data is only analyzed once SS is reached? Is there valuable data in the first 5 hours? If the HOM ratios are changing during this build-up time there should be further clues to the HOM formation mechanism? In the Helsinki experiments you have not reached SS, so where along this HOM concentration curve is applicable? DOES ANYONE HAVE A KINETIC MODEL TO EXPLAIN FIGURE 2?

R21. It is true that we limit our analysis to steady-state conditions. This is mainly because when lights are turned on we have a fast new particle formation and fast changes in ozone, OH, HOM and particle concentrations. In addition, modelling of gas-particle partitioning would require the knowledge of lower oxygenated compounds, information we do not have. Therefore, we would not be able to disentangle these different effects on HOM formation before steady state is reached. The time to reach SS in JPAC depended on how many particles were formed, and in high VOC experiments, it typically took ~5h. We cannot directly put the result of the flow reactor to the chamber HOM curve (if reviewer means Figure 2) as the both the reaction system and time scales are different (initial OH in flow reactor that reacts away versus OH produced constantly in JPAC) different.

In Equation 5, γ is in the denominator.
R22. We fixed the equation.

page 15, 4 "for each HOM molecule i" Can you indicate the value of i, i.e. how many HOMs are considered?
R23. Please see response R6.

page 16, 8 "HO2 can also be produced in our reactor making a reaction with HO2 an important bimolecular termination pathway in our system." Please make in clear in the experimental that HO2 is made from H2O photolysis.
R24. We added this to the Methods:
"Hydroxyl radicals were produced via photolysis of water at 184.9nm, a reaction that also caused the co-production of $HO_2$"

Is there any specific reason that 185 nm was used to generate OH. If 254 nm and O3/H2O was used then there would be a more direct comparison to the JPAC experiment. Can the lamp be translated along the reactor to change the contact time? I note that Molteni(2018) used an excimer lamp at 172 nm. Both 185nm and 172nm will photolyze aromatics, but this is not the case for 254 nm.

R25. We performed the flow reactor experiments before JPAC, and with our existing hardware, we were limited to the 184.9nm light source.

The results in Figure 3 show that a range of HOMs are formed and the text provides some explanation of the type of reactions required to make each HOM. I was wondering if this could be quantified further. For instance, the dimers require an RO2 + RO2 bimolecular reaction. What sort of rate coefficient is required? While you state you do not know OH concentrations in the Helsinki experiments this makes it a difficult question, but in the JPAC experiments maybe this is possible.

R26. In JPAC experiments, the estimation of rate coefficients would be impossible primarily for two reasons: firstly, the long residence time and the fact that we cannot distinguish between $RO_2$ produced in different generations of OH oxidation; and secondly, a given ROOR likely has many isomers, and each $RO_2 + RO_2$ pair will have different reaction rates (as shown e.g. by Berndt et al. 2018).

page 18, 13 "The OH production from H2O photolysis stayed constant in our experiments, but the VOC acts as a sink for the OH radicals, which means that higher VOC concentrations will result in lower OH concentrations." Is there a reason why benzene or toluene was not lowered to maybe promote 2 OH reactions? If you had the bubbler before the MFC would you have greater control of the [VOC]?

R27. The main reason is because flow reactor experiments were conducted before JPAC, where we realized the effect of multi-generation oxidation on the formation of HOM. In addition, we operated at the smallest possible flows of MFC, 1 mlpm, meaning smallest VOC concentrations possible. On the other hand, it would be very difficult to generate higher OH in such a short time scale system without triggering other reactions that would obscure the multigeneration oxidation (for example, $RO_2$+OH recently receiving considerable attention).

Since, MFC requires a few bars pressure, placing a glass bubbler before the MFC would pressurize the bubbler.

In the Helsinki experiments, if the residence time could be changed then the mechanism of how these HOMs are formed might be clearer and even provide some kinetic assignments. If the lamp was located further along the reactor would this significantly change the residence time?

R28. As mentioned before, at the time of flow reactor studies we did not realise all the relevant aspects of HOM formation in aromatic system. The effect of residence time would be very useful in order to elucidate more exact mechanistic details of the oxidation reactions, but is outside the scope of this work, where we focus on the effect of multi-step OH oxidation on HOM formation.

Page 18, 31 "At small oxidation rates, the total HOM concentration increased linearly, but reached a plateau around 3-4 $\times$ 10$^8$ cm$^{-3}$ at higher oxidation rate." This plateau is true for the light green square but not for the dark blue!

R29. As discussed later in the same section, plateau is reached for certain high VOC experiments (lower OH, lighter green in Figure 4a). This is due to the high amount of particles formed in those experiments.

We rephrased the sentence:

"In benzene experiments at small oxidation rates, the total HOM concentration increased linearly, but a plateau around 3 $\times$ 10$^8$ cm$^{-3}$ was visible at higher oxidation rate."
"However, especially in the high [VOC] experiments (markers on the right hand side of Fig. 4a), the CS

due to particles formed in the chamber was of the same order as the wall loss and thus the approximation that $k_{loss}$ equals $k_{wall}$ is not valid anymore."

From Figure 4 b, HOM yield appears to be linear in OH. Surely there must be a limit where the HOM yield plateaus? If the HOM yield is increasing with more multiple OH reactions, does this mean that HOMmulti ⇒ particles is slower?

R30. We would like to point out that x-axis in Figure 4b is in logarithmic scale and, therefore, the yield dependence on OH is not linear. It is definitely the case that HOM yield will reach a plateau at some OH concentration. We, however, did not reach such a condition in our experiments.

"HOMmulti ⇒ particles is slower" – we assume the reviewer means that the formation rate of SOA would be slower because multiple oxidation steps must take place before the products become condensable. This delay in SOA formation has been observed in high NOx benzene oxidation experiments by Ng et al. (2007), but no delay was seen in no-NOx SOA experiments. In our SOA experiments, we would not be able to observe a delay because we introduced seed aerosols when the system was in steady state, while Ng et al. (2007) switched the oxidation on when seed concentration reached steady state. The HOM condensation and SOA formation at atmospherically-close concentration is highly non-linear especially as seed aerosol is in competition with wall loss. In addition, the rate of SOA formation will depend largely on how much of seed aerosol is added and what is the total concentration of condensable vapours and may not directly depend on the yield of HOM unless it is a nucleation SOA experiment (in absence of seed).

Page 19, 11 "Our estimated HOM yields from benzene oxidation were 4.1-14.0%, which can be compared to a value of 0.2% provided by Molteni et al. (2018). Their value likely corresponds to the HOM yield of the first OH oxidation step, potentially also impacted by a second step, suggesting that more than 90% of the "HOM- forming potential" of benzene comes from multi-generation OH oxidation." I'm confused here as to which study is relevant to the atmosphere. Molteni(2018) had a factor of ten less HOM than the present study, is this consistent with the OH concentration they used in their experiment? Which HOM yield would go into a model? Can you provide more discussion in the paper about this difference? Have you artificially raised the [OH] in order to bring about large [HOM]?

R31. Please refer to response R1 and R17.

The key differences between flow reactor (as in Molteni et al.) and continuous flow chamber studies (as in JPAC) is that in the former, the air parcel is provided with an initial OH concentration, which then reacts away as the parcel moves down the reactor. In the continuous flow chamber, however, we constantly produce OH during the 48 min residence time of the air inside. Therefore, OH concentrations cannot be directly compared. Instead, OH dose would be more appropriate value as it also takes into account the residence time. The initial concentration of OH in Molteni et al. was $8.5 \times 10^{11}$ cm$^{-3}$ and VOC concentration was about 4ppm, while in our study OH was $10^7$-$5 \times 10^8$ cm$^{-3}$ and VOC concentration was 1-100 ppb. Technically, the concentrations in our experiments were closer to the atmospheric values, but in terms of studying HOM yields, both studies are relevant, as the result is a function of reaction timescales.

"Have you artificially raised the [OH] in order to bring about large [HOM]?" – We conducted the experiments, in which we did produce OH artificially, with the aim of seeing the response at higher OH dose.

If you looked at the HOMi signal at early times, the 5 hours before steady-state, can you see the various HOMi evolving in time?

R32. Please refer to response R21.

Now if you are saying that the HOM yield is 10 times smaller for 1 OH reaction, then should the HOM spectra dramatically change, i.e. big difference between Figure 3 and 5?

R33. We discuss this difference in HOM composition in section 4.4.2. There are clear differences between Figure 3a and Figure 5a-c. The change is not very dramatic since the composition of secondary products, such as phenol and chatechol, have similar C and H content as benzene; however, the change is significant.

Page 21, 1 "However, after this, OH oxidation can only proceed via H-abstraction, and if the subsequent termination reactions occur by loss of OH or HO2, a decrease in H-atoms will take place. In other words, it is to be expected, that multi-generation OH oxidation will produce also molecules with fewer H-atoms than the parent VOC." Are these high [OH] HOM relevant to the atmosphere?

R34. If these intermediates, precursors for HOM, are semi-volatile, they may remain in the gas-phase in the atmosphere long enough to react several times with OH, which is equivalent to our high [OH]. A given VOC molecule receives in our experiments the same OH dose as a VOC molecule in the atmosphere over a course of 10 hours to 15 days. Therefore, the observed HOM are expected to be relevant to the atmosphere. Please also see response R17 in relation to OH dose.

Page 21, 3 "Another possibility is that the dimer formation upon RO2+R'O2 reaction would be less likely for the RO2 formed at high OH." I would have thought RO2 + RO2 is more likely the higher the OH concentration.

R35. Here we discuss the ratio between dimers and monomers. The formation of dimers will depend on RO2 concentration as well as on the dimer formation rate constant. If at high OH concentration, the RO2 structure is such that the dimer formation rate constant for is lower, then it could explain the observed decrease in dimer-to-monomer ratio. We clarified the sentence:

"This may be explained by higher HO2 concentrations at higher OH. Another possible explanation is that RO2 formed at higher OH would have less favourable structures for dimer formation. The dimer formation rate has been shown to be highly dependent on the structure of the reacting RO2 (Berndt et al., 2018a)".

Page 23, 22 "but also for some precursors for multi-generation HOM formation that are undetected by our instrument (or detected at lower sensitivity)." But I thought you are saying that the multi-generation HOM are from the primary generation HOM.
OH + HOMprimary ⇒ HOMmulti
What sort of intermediate before HOMmulti do you think might be present that is not detected? I understand why BPR is not detected but not what you are presently suggesting.

R36. We have realized that using the term "multi-generation HOM" could be misleading and changed it to "HOM formed in multi-generation OH oxidation". The specific sentence was changed as follows:

"but also for some of the undetected oxidation products (or detected at low sensitivity), that could have formed detectable HOM upon further OH oxidation steps. This explanation is plausible and is in support of our hypothesis that some of the HOM were formed in multi-generation OH oxidation."

If HOM formed in multiple OH oxidation steps was from other HOM that we could observe, we would not observed an increase in HOM yield. The increase in HOM molar yield tells us that some HOM were formed from lower-oxygenated compounds that we cannot detect (e.g. phenol, catechol, etc) or from those that we detect with lower sensitivity, or both. Currently, it is impossible to tell the following oxidation steps of which compounds form HOM. From Hyttinen et al. (2015), we know that even if a compound has 6 oxygen atoms and two H-donor functional groups, it does not mean it will necessarily be charged at collision limit in our chemical ionization inlet.

In section 4.2.1 we have included a short clarification what we mean under HOM formed in multi-generation OH oxidation:

"Since the observed HOM molar yield increased, we can conclude that the undetected lower oxygenated products reacted again with OH to form more of the detectable HOM. These intermediate precursors could also be higher oxygenated compounds that were detected in our instrument with ionisation efficiency below the collision limit (Hyttinen et al. 2015).
…
To test secondary OH oxidation, we conducted three similar experiments starting with phenol as the precursor, a known first-generation oxidation product of benzene."

Page 24, 30 "In addition, we conclude that atmospheric models should take into account HOM yield dependence on the chemical regime when implementing quantitative laboratory results." Can you make it clear which study is relevant to the atmosphere, the present work (high HOM yield) or Molteni(2018) (low HOM yield).

R37. Please refer to response R1.

References:
Berndt, T., Mentler, B., Scholz, W., Fischer, L., Herrmann, H., Kulmala, M., and Hansel, A.: Accretion Product Formation from Ozonolysis and OH Radical Reaction of α-Pinene: Mechanistic Insight and the Influence of Isoprene and Ethylene, Environmental Science & Technology, 52, 11 069–11 077, https://doi.org/10.1021/acs.est.8b02210, 2018.

Chen, T., Liu, Y., Chu, B., Liu, C., Liu, J., Ge, Y., Ma, Q., & Ma, J., and He, H.: Differences of the oxidation process and secondary organic aerosol formation at low and high precursor concentrations. Journal of Environmental Sciences. 79. 10.1016/j.jes.2018.11.011, 2018.

Hammes, J., Tsiligiannis, E., Mentel, T. F., and Hallquist, M.: Effect of NOx on 1,3,5-trimethylbenzene (TMB) oxidation product distribution and particle formation, Atmos. Chem. Phys. Discuss., 2019, 1-18, 10.5194/acp-2019-395, 2019.

Hyttinen, N., Kupiainen-Määttä, O., Rissanen, M. P., Muuronen, M., Ehn, M., and Kurtén, T.: Modeling the Charging of Highly Oxidized Cyclohexene Ozonolysis Products Using Nitrate-Based Chemical Ionization, The Journal of Physical Chemistry A, 119, 6339–6345, https://doi.org/10.1021/acs.jpca.5b01818, 2015.

McFiggans, G., Mentel, T. F., Wildt, J., Pullinen, I., Kang, S., Kleist, E., Schmitt, S., Springer, M., Tillmann, R., Wu, C., Zhao, D., Hallquist, M., Faxon, C., Le Breton, M., Hallquist, Å. M., Simpson, D., Bergström, R., Jenkin, M. E., Ehn, M., Thornton, J. A., Alfarra, M. R., Bannan, T. J., Percival, C. J., Priestley, M., Topping, D., and Kiendler-Scharr, A.: Secondary organic aerosol reduced by mixture of atmospheric vapours, Nature, 565, 587-593, 10.1038/s41586-018-0871-y, 2019.

Molteni, U., Bianchi, F., Klein, F., Haddad, I. E., Frege, C., Rossi, M. J., Dommen, J., and Baltensperger, U.: Formation of highly oxygenated organic molecules from aromatic compounds, Atmospheric Chemistry and Physics, 18, 1909–1921, https://doi.org/10.5194/acp-18-1909-2018, 2018.

Ng, N. L., Kroll, J. H., Chan, A.W. H., Chhabra, P. S., Flagan, R. C., and Seinfeld, J. H.: Secondary organic aerosol formation fromm-xylene, toluene, and benzene, Atmospheric Chemistry and Physics, 7, 3909–3922, https://doi.org/10.5194/acp-7-3909-2007, 2007.

Wang, S., Wu, R., Berndt, T., Ehn, M., and Wang, L.: Formation of Highly Oxidized Radicals and Multifunctional Products from the Atmospheric Oxidation of Alkylbenzenes, Environmental Science&Technology, 51, 8442–8449, https://doi.org/10.1021/acs.est.7b02374, 2017.